**Article** https://doi.org/10.1038/s41467-023-42181-x

# Future precipitation increase constrained by climatological pattern of cloud effect

Wenyu Zhou [1] ✉, L. Ruby Leung [1], Nicholas Siler [2] & Jian Lu [1]

The fractional increase in global mean precipitation ($\triangle \bar{P}/\bar{P}$) is a first-order measure of the hydrological cycle intensification under anthropogenic warming. However, $\triangle \bar{P}/\bar{P}$ varies by a factor of more than three among model projections, hindering credible assessments of the associated climate impacts. The uncertainty in $\triangle \bar{P}/\bar{P}$ stems from uncertainty in both hydrological sensitivity (global mean precipitation increase per unit warming) and climate sensitivity (global mean temperature increase per forcing). Here, by investigating hydrological and climate sensitivities in a unified surface-energy-balance perspective, we find that both sensitivities are significantly correlated with surface shortwave cloud feedback, which is further linked to the climatological pattern of cloud shortwave effect. The observed pattern of cloud effect thus constrains both sensitivities and consequently constrains $\triangle \bar{P}/\bar{P}$. The 5%-95% uncertainty range of $\triangle \bar{P}/\bar{P}$ from 1979-2005 to 2080-2100 under the high-emission (moderate-emission) scenario is constrained from $6.34 \pm 3.53\%$ ($4.19 \pm 2.28\%$) in the raw ensemble-model projection to $7.03 \pm 2.59\%$ ($4.63 \pm 1.71\%$). The constraint thus suggests a higher most-likely $\triangle \bar{P}/\bar{P}$ and reduces the uncertainty by ~25%, providing valuable information for impact assessments.

Global mean precipitation ($\bar{P}$) is projected to increase under anthropogenic warming, but the magnitude of increase is highly uncertain. The fractional increase in mean precipitation over the 21st century ($\triangle \bar{P}/\bar{P}$) varies from 3.3 to 11.6% in models under the high-emission scenario[1-3] (RCP8.5/SSP5-8.5) and from 1.9 to 7.4% under the moderate-emission scenario (RCP4.5/SSP2-4.5). This large uncertainty in $\triangle \bar{P}/\bar{P}$ must be constrained for credible assessments of hydrological impacts. The uncertainty in $\triangle \bar{P}/\bar{P}$ stems from uncertainty in both hydrological sensitivity (HS) which represents the fractional increase in global mean precipitation per degree of warming ($\triangle \bar{P}/\bar{P}/\triangle \bar{T}$) and climate sensitivity (CS) which represents the increase in global mean temperature per anthropogenic forcing ($\triangle \bar{T}$). CS and HS contribute roughly equally to uncertainty in global precipitation change, both varying by a factor of two among models (ratio between the highest and lowest values)[4,5]. An accurate estimate of future precipitation increase thus requires constraining uncertainty in both HS and CS.

A common approach to constraining uncertainty in climate-model projections is the so-called "emergent constraint" (EC), which utilizes statistical and/or physical relationships between the observable current climate and a future projection[6,7]. For CS, several ECs have been proposed from various perspectives[8], such as linking future temperature change to its seasonal[9] or interannual variability[10], relating cloud feedbacks to the climatological distribution[11,12] or properties[13,14] of clouds, and using correlations with other observable features[15,16]. For HS, two ECs have been proposed[17,18] based on the atmosphere-energy-balance perspective that relates changes in global precipitation to changes in atmospheric radiative cooling[1,19,20]. They show that HS is correlated with feedbacks from clear-sky shortwave absorption[17] and longwave cloud radiation[18]. However, a more recent analysis[21] finds that, in the latest generation of climate models, the correlation of HS to clear-sky shortwave feedback weakens substantially (r = −0.45) and the correlation to longwave cloud feedback no longer holds (r=-0.12). Furthermore, the constraints on HS and CS have not been integrated into a full constraint on $\triangle \bar{P}/\bar{P}$. It was challenging to do so as the ECs on HS and CS were based on different but potentially correlated observables.

[1]Atmospheric, Climate, and Earth Sciences Division, Pacific Northwest National Laboratory, Richland, WA, USA. [2]College of Earth, Ocean, and Atmospheric Sciences, Oregon State University, Corvallis, OR, USA. ✉e-mail: wenyu.zhou@pnnl.gov

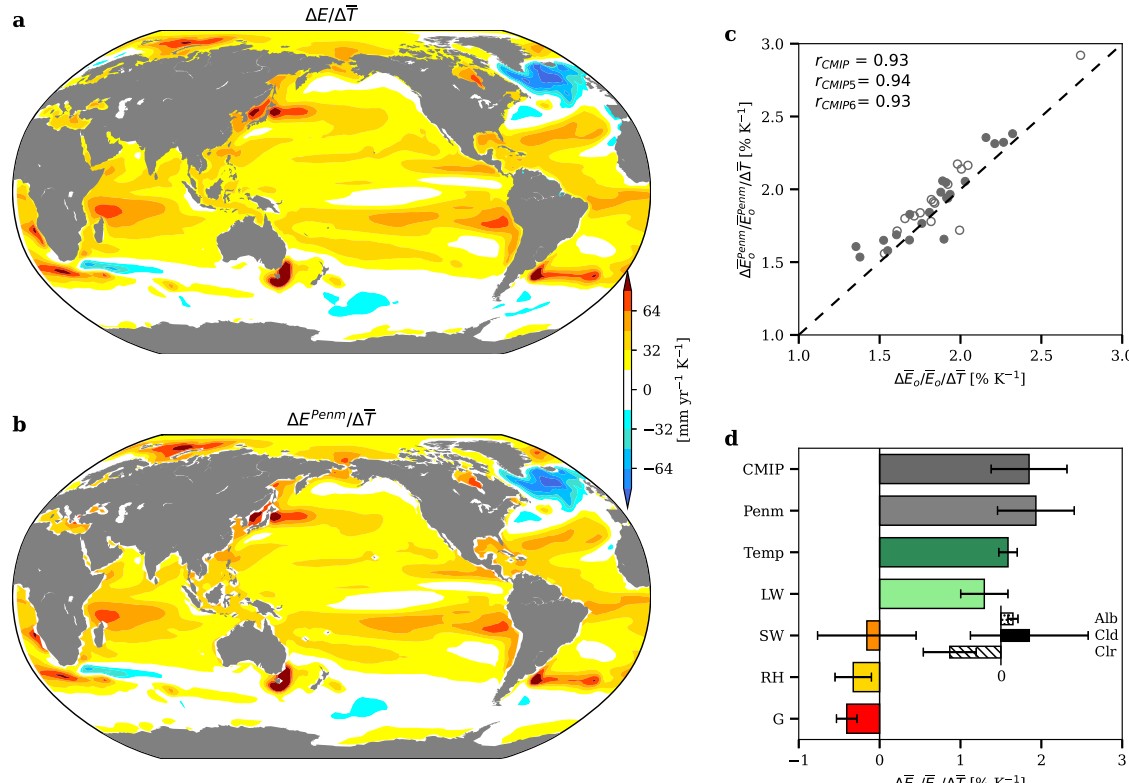

**Fig. 1 | Attribution of future ocean evaporation change and its uncertainty to different factors. a** Spatial pattern of ocean evaporation changes per unit warming projected by models ($\Delta E/\Delta\bar{T}$). **b** Spatial pattern of ocean evaporation changes per unit warming estimated by the Penman equation ($\Delta E^{Penm}/\Delta\bar{T}$). **c** Intermodel scatterplot between the model-projected and Penman-estimated increases in global mean ocean evaporation per unit warming. The open and filled dots are the CMIP5 and CMIP6 results, respectively. The intermodel correlation, $r$, is computed for separated and combined CMIP ensembles. **d** Contributions of individual factors to the ensemble model mean and uncertainty of global mean ocean evaporation increase per unit warming ($\Delta\bar{E}_o/\bar{E}_o/\Delta\bar{T}$). The contribution of surface shortwave flux is further decomposed into that from changes in albedo, clear-sky downward shortwave flux and cloud effect on downward shortwave flux. The results are based on RCP8.5/SSP5-8.5 and projected changes are computed based on the difference between the historical (1979–2005) and future (2080–2100) periods.

Using a rather different approach, some studies have directly constrained $\Delta\bar{P}/\bar{P}$ by relating it to the observed historical temperature trend[1,22]. The rationale is that future precipitation increase should be correlated with future temperature increase and future temperature increase should be correlated with forced temperature trend over recent decades. Interestingly, the correlation of $\Delta\bar{P}/\bar{P}$ to historical precipitation trend is lower and the resultant EC is less effective. Thus, their ECs mainly utilized the CS part while the potential to constrain the HS part was not harnessed. Furthermore, as the observed trend contains not only the forced response but also natural variability, the forced temperature trend must be estimated as a range, which introduces further uncertainty.

Is it possible to build an EC on $\Delta\bar{P}/\bar{P}$ that integrates constraints on both CS and HS? Here, unlike previous studies that consider HS and CS separately from the perspectives of atmosphere and top-of-the-atmosphere, we investigate them in a unified surface-energy-balance perspective. This allows us to correlate CS and HS with the same observable and upon that build an EC on $\Delta\bar{P}/\bar{P}$. Specifically, we reveal that the sensitivity of surface shortwave cloud effect to anthropogenic warming is responsible for a large part of the uncertainty in both HS and CS. We further show that the uncertainty in cloud sensitivity is dominated by the temperature-mediated feedback and can be inferred from the climatological pattern of cloud shortwave effect. The observed pattern of cloud shortwave effect thus provides an effective EC on $\Delta\bar{P}/\bar{P}$. The resultant EC indicates a higher most-likely $\Delta\bar{P}/\bar{P}$ than the ensemble model projection and reduces the 5%-95% uncertainty range of $\Delta\bar{P}/\bar{P}$ by ~25%.

Our analyses are based on future projections of climate models in Coupled Model Intercomparison Project (CMIP)[23,24] phases 5 and 6.

The main results focus on the high-emission scenario (RCP8.5/SSP5-8.5) but consistent results are found in the medium-emission scenario (RCP4.5/SSP2-4.5). Future changes are computed as the change in climatology from the historical (1979-2005) to future (2080-2100) period. The robustness of the result is shown in both individual and combined ensembles of CMIP5 and CMIP6.

## A surface-energy-balance perspective for understanding HS and its uncertainty

Because the increase in global precipitation is strongly correlated with the increase in global ocean evaporation (Fig. S1), one can understand HS from the perspective of ocean surface energy balance[25,26]. Specifically, according to a reformulated Penman equation[25,27], ocean evaporation is linked to surface energy fluxes and relative humidity as,

$$L_v E_o^{Penm} = f[SW + LW - G + \xi(1 - RH)]$$

where $SW$ and $LW$ are the net shortwave and longwave radiative fluxes received by ocean surface; $G$ is the ocean heat uptake; $RH$ is the near-surface relative humidity; $f$ is an energy allocation factor that depends only on temperature and increases with warming (reflecting an increased ratio of latent to sensible surface fluxes); and $\xi$ is a scale parameter that varies with aerodynamic resistance. A derivation with detailed definitions of $f$ and $\xi$ is provided in Methods. The Penman equation allows us to attribute future change in ocean evaporation and its intermodel uncertainty to individual factors.

We first affirm that the Penman equation well captures the historical climatology (Fig. S2) and future change (Fig. 1) of the model-

simulated ocean evaporation. The spatial patterns of the estimated and simulated evaporation are correlated at r = 0.99 for historical climatology (Fig. S2a–c) and r = 0.98 for future change (Fig. 1a, b). Across models, the global mean values of the estimated and simulated evaporation are correlated at r = 0.78 for historical climatology (Fig. S2d) and r = 0.93 for future change (Fig. 1c).

We then attribute future change in ocean evaporation and its uncertainty to individual factors such as surface air temperature (which controls the energy allocation factor $f$), surface radiative fluxes, relative humidity, ocean heat uptake, and surface wind speed (which affects aerodynamic resistance and then the parameter $\xi$). As summarized in Fig. 1d, the model ensemble projects a 1.87 ± 0.46 % K$^{-1}$ increase in global ocean evaporation ($\triangle\bar{E}_o/\bar{E}_o/\triangle\bar{T}$), where ± indicates the 5–95% uncertainty range (Methods). The increase is contributed by the warmer temperature that increases the energy allocation factor $f$[25,28] and the increasing net longwave flux as a more humid atmosphere enhances air emissivity[29], but is partially offset by the ocean heat uptake and relative humidity increase (see Fig. S3 for the spatial patterns of changes in individual factors and their contributions to ocean evaporation change). While surface shortwave flux only contributes weakly to the mean change, it is the largest source of uncertainty, leading to a ± 0.61 % K$^{-1}$ intermodel spread in global ocean evaporation change. The contribution of surface shortwave flux is further decomposed into contributions from surface albedo, clear-sky downward shortwave flux, and cloud effect on downward shortwave flux (hereafter denoted as surface shortwave cloud effect, or SSCE). Of these, SSCE is by far the largest source of the uncertainty, leading to a ± 0.72 % K$^{-1}$ spread in global ocean evaporation change (Fig. 1d; see Fig. S4 for the contribution of individual factors in each model). Note that the ± 0.72% K$^{-1}$ uncertainty due to the SSCE change is larger than the total 0.46% K$^{-1}$ uncertainty in HS, indicating potential compensations from other factors. In particular, the intermodel uncertainty due to SSCE is partially (-0.3) compensated by that due to surface net longwave radiation (Fig. S5).

## Surface shortwave cloud feedback regulates future precipitation change by affecting both CS and HS

Due to the dominant contribution of SSCE to the projection uncertainty, the increase in global ocean evaporation per unit warming ($\triangle\bar{E}_o/\bar{E}_o/\triangle\bar{T}$) is significantly correlated with that contributed solely by the SSCE change ($\triangle\bar{E}_o^{SSCE}/\bar{E}_o/\triangle\bar{T}$) across models (Fig. S6; r = 0.75). As a result, $\triangle\bar{E}_o/\bar{E}_o/\triangle\bar{T}$ is significantly correlated with global ocean SSCE change per unit warming ($\triangle\overline{SSCE}/\triangle\bar{T}$) (Fig. 2a; r = 0.72). Models with a larger $\triangle\overline{SSCE}/\triangle\bar{T}$ tend to project a larger $\triangle\bar{E}_o/\bar{E}_o/\triangle\bar{T}$, that is, a higher HS in terms of ocean evaporation increase.

According to surface energy balance, a larger $\triangle\overline{SSCE}/\triangle\bar{T}$ would also enhance the degree of surface temperature warming under anthropogenic forcing ($\triangle\bar{T}$), implying a potential positive correlation between $\triangle\overline{SSCE}/\triangle\bar{T}$ and CS. Indeed, CS is significantly correlated with $\triangle\overline{SSCE}/\triangle\bar{T}$ across models (Fig. 2b; r = 0.61). Models with a larger $\triangle\overline{SSCE}/\triangle\bar{T}$ tend to project a larger global mean surface warming $\triangle\bar{T}$.

As both $\triangle\bar{E}_o/\bar{E}_o/\triangle\bar{T}$ and $\triangle\bar{T}$ are correlated with $\triangle\overline{SSCE}/\triangle\bar{T}$, $\triangle\bar{E}_o/\bar{E}_o$ (Fig. 2c; r = 0.76) and thus $\triangle\bar{P}/\bar{P}$ (Fig. 2d; r = 0.70) are correlated with $\triangle\overline{SSCE}/\triangle\bar{T}$. All these correlations are significant among both the individual and combined ensemble of CMIP5 and CMIP6 (see correlation coefficients in Fig. 2). The high correlation between $\triangle\overline{SSCE}/\triangle\bar{T}$ and $\triangle\bar{P}/\bar{P}$ suggests that if there is an observational constraint on $\triangle\overline{SSCE}/\triangle\bar{T}$, we can also constrain the uncertainty in $\triangle\bar{P}/\bar{P}$ under anthropogenic warming.

$\triangle\overline{SSCE}/\triangle\bar{T}$ is the change in global ocean SSCE per unit warming under the RCP8.5/SSP5-8.5 scenario, which includes the effects of both rapid adjustment and temperature-mediated feedback (feedback here refers to the temperature-mediated change in SSCE, without corrections for non-cloud factors as in the radiative-kernel method). By diagnosing these two effects from the piControl and Abrupt4xCO2

experiments (Methods), we can see that the effect of rapid adjustment of SSCE (normalized by ocean warming) is much less uncertain compared to the temperature-mediated feedback of SSCE, hereafter referred to as SSCF (Fig. 2e). As a result, the intermodel uncertainty in $\triangle\overline{SSCE}/\triangle\bar{T}$ is largely due to that of SSCF, with the intermodel correlation between $\triangle\overline{SSCE}/\triangle\bar{T}$ and SSCF at r = 0.9 (Fig. 2f). Subsequently, $\triangle\bar{E}_o/\bar{E}_o/\triangle\bar{T}$, $\triangle\bar{T}$, $\triangle\bar{E}_o/\bar{E}_o$ and $\triangle\bar{P}/\bar{P}$ are all significantly correlated with SSCF (Fig. S7).

## An emergent constraint on future precipitation increase

Now, we explore potential correlations of $\triangle\overline{SSCE}/\triangle\bar{T}$ to observable quantities and build ECs for $\triangle\overline{SSCE}/\triangle\bar{T}$ and $\triangle\bar{P}/\bar{P}$. The shortwave cloud radiative effect and its feedback have been widely studied from the top-of-the-atmosphere (TOA) perspective[30–32]. Previous work finds that the intermodel uncertainty in TOA shortwave cloud feedback can be attributed to distinct climatological patterns of cloud effect[12]. Particularly, models that exhibit lower (higher) climatological cloud reflection in warm (cool) regions tend to project a more positive shortwave cloud feedback at the TOA. Because the uncertainty in $\triangle\overline{SSCE}/\triangle\bar{T}$ is dominated by the temperature-mediated feedback and both shortwave cloud effect and feedback are highly correlated between surface and TOA (Fig. S8), we explore a similar correlation between $\triangle\overline{SSCE}/\triangle\bar{T}$ and the climatological pattern of SSCE.

To visualize the potential relation between $\triangle\overline{SSCE}/\triangle\bar{T}$ and the climatological pattern of SSCE, we compare model groups with the highest and lowest $\triangle\overline{SSCE}/\triangle\bar{T}$ (Fig. 3). The two groups present distinct climatological patterns of SSCE and local $\triangle SSCE/\triangle\bar{T}$. For the climatological SSCE pattern, the group of high $\triangle\overline{SSCE}/\triangle\bar{T}$ exhibits a larger spatial variance of SSCE, that is, stronger SSCE (more negative than that in an average model) over the cold midlatitudinal region where SSCE is climatologically strong and weaker SSCE over the warm tropical region (except the deep connective regions) where SSCE is climatologically weak (Fig. 3a, c, e). For local $\triangle SSCE/\triangle\bar{T}$, the group of high $\triangle\overline{SSCE}/\triangle\bar{T}$ exhibits more positive $\triangle SSCE/\triangle\bar{T}$ over subtropical regions where the climatological SSCE exhibits strong spatial gradients (Fig. 3b, d, f). The relation between $\triangle\overline{SSCE}/\triangle\bar{T}$, local $\triangle SSCE/\triangle\bar{T}$, and climatological SSCE can be interpreted similarly as those at the TOA[12]. Specifically, as the warm tropical region with lower climatological SSCE expands poleward with warming[33,34], a larger spatial gradient in the climatological SSCE induces an anomalously positive $\triangle SSCE/\triangle\bar{T}$ in the subtropical transition regions (Fig. 3f), consequently leading to a higher $\triangle\overline{SSCE}/\triangle\bar{T}$.

A consistent picture is seen from the regression of the climatological SSCE onto $\triangle\overline{SSCE}/\triangle\bar{T}$ across models. A higher $\triangle\overline{SSCE}/\triangle\bar{T}$ is associated with lower (less negative) climatological SSCE over the warm tropical regions and higher (more negative) climatological SSCE over cold midlatitudinal regions (Fig. 4a). Thus, an index that measures how much a model climatology favors high $\triangle\overline{SSCE}/\triangle\bar{T}$ can be derived as the pattern correlation between the climatological SSCE (anomaly relative to the multi-model mean) and Fig. 4a. This index of climatological SSCE pattern, denoted as $C$, is significantly correlated with $\triangle\overline{SSCE}/\triangle\bar{T}$ across models (r = 0.72; Fig. 4b). The ISCCP[35] and CERES[36] datasets provide satellite measurements of surface radiative flux and their climatological values of Index $C$ are 0.25 and 0.28, respectively. An EC on $\triangle\overline{SSCE}/\triangle\bar{T}$ is then derived by applying the observational values of $C$ to the regression line. The resultant EC constrains the 5%-95% uncertainty range of $\triangle\overline{SSCE}/\triangle\bar{T}$ from 0.39 ± 1.20 W m$^{-2}$ K$^{-1}$ in the raw ensemble-model projection (black vertical error bar in Fig. 4b) to 0.63 ± 0.83 W m$^{-2}$ K$^{-1}$ (red vertical error bar in Fig. 4b).

As $\triangle\bar{P}/\bar{P}$ is significantly correlated with $\triangle\overline{SSCE}/\triangle\bar{T}$ and $\triangle\overline{SSCE}/\triangle\bar{T}$ is significantly correlated with Index C, $\triangle\bar{P}/\bar{P}$ is significantly correlated with Index C (Fig. 4c; r = 0.68). We then derive an EC on $\triangle\bar{P}/\bar{P}$ by applying the observational values of $C$ to the regression line between $C$ and $\triangle\bar{P}/\bar{P}$. The resultant EC constrains the 5%-95% uncertainty range of $\triangle\bar{P}/\bar{P}$ between 1979-2005 and 2080-2100 under

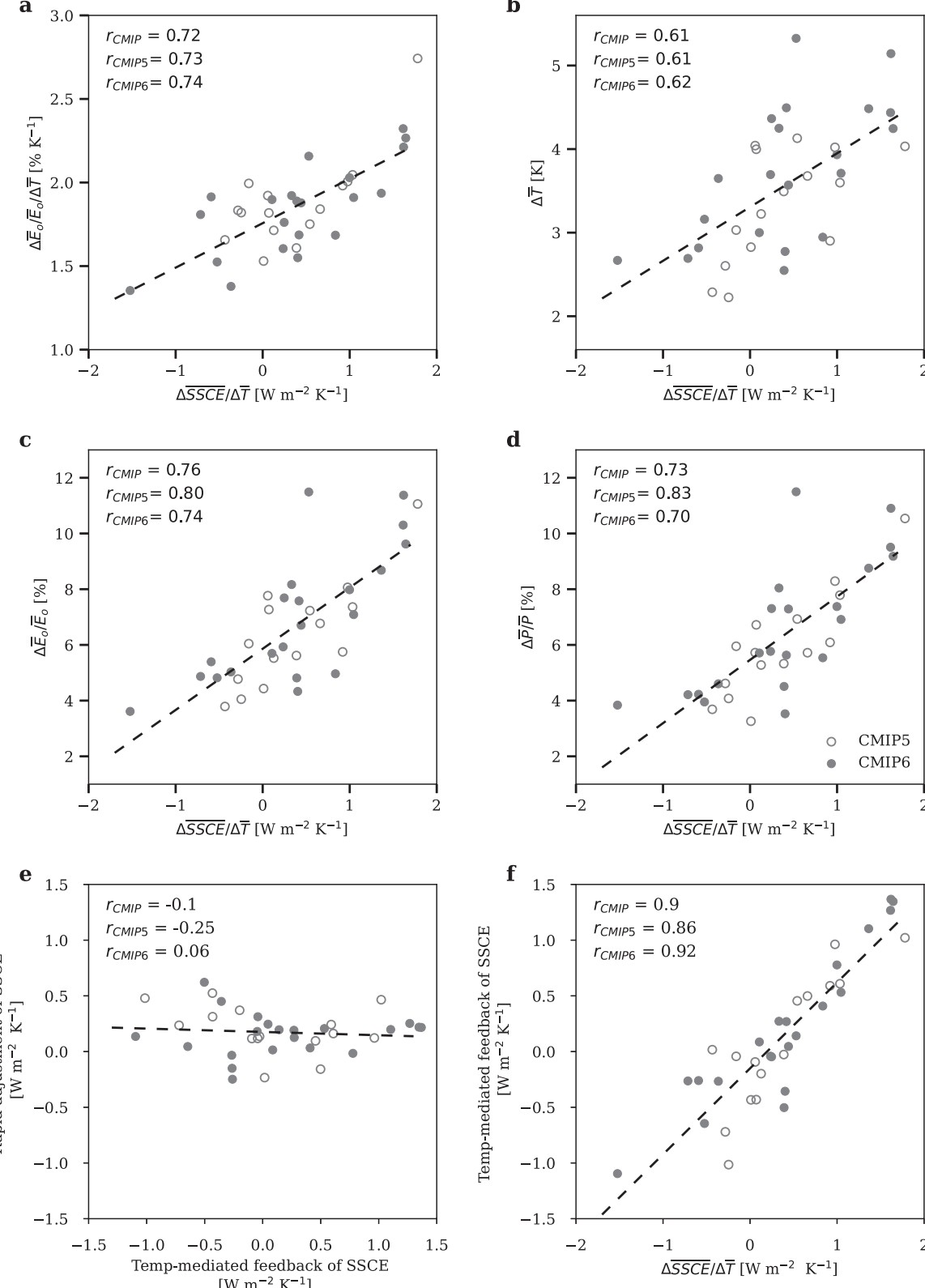

**Fig. 2 | Relating hydrological sensitivity (HS) and climate sensitivity (CS) to the sensitivity of global ocean surface shortwave cloud effect to warming ($\Delta\overline{SSCE}/\Delta\overline{T}$).** **a** Intermodel scatterplot between $\Delta\overline{SSCE}/\Delta\overline{T}$ and HS in terms of ocean evaporation ($\Delta\overline{E}_o/\overline{E}_o/\Delta\overline{T}$). **b** Intermodel scatterplot between $\Delta\overline{SSCE}/\Delta\overline{T}$ and CS ($\Delta\overline{T}$). **c** Intermodel scatterplot between $\Delta\overline{SSCE}/\Delta\overline{T}$ and global mean ocean evaporation increase ($\Delta\overline{E}_o/\overline{E}_o$). **d** Intermodel scatterplot between $\Delta\overline{SSCE}/\Delta\overline{T}$ and global mean precipitation increase ($\Delta\overline{P}/\overline{P}$). **e** Intermodel scatterplot between rapid adjustment and temperature-mediated feedback of SSCE diagnosed from Abrupt4xCO2. **f** Intermodel scatterplot between $\Delta\overline{SSCE}/\Delta\overline{T}$ in RCP8.5/SSP5-8.5 and temperature-mediated feedback of SSCE diagnosed from Abrupt4xCO2. The open and filled dots are the CMIP5 and CMIP6 results, respectively. The intermodel correlation, r, is computed for all the models, the CMIP5 models only and the CMIP6 models only.

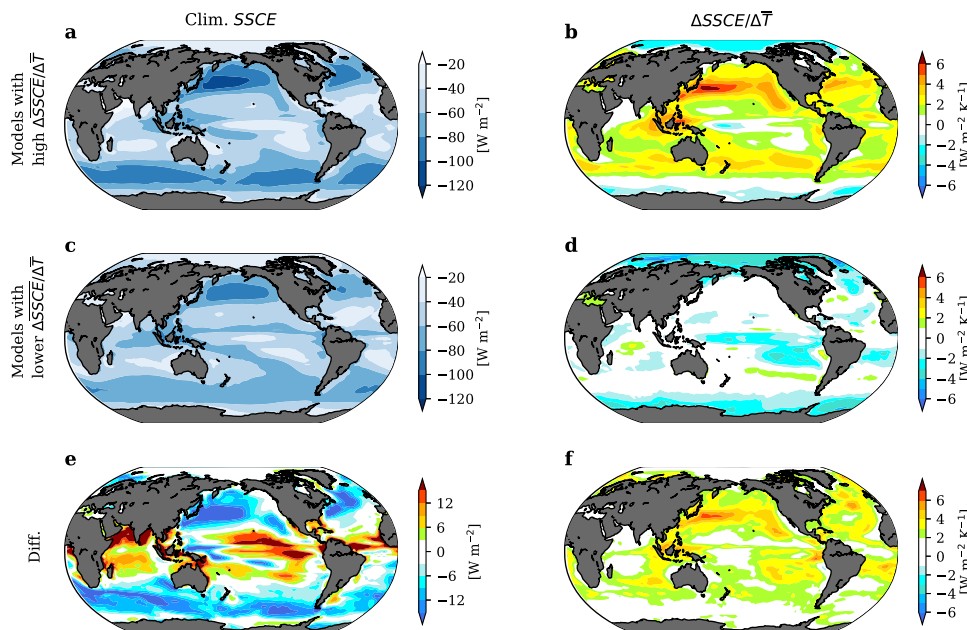

**Fig. 3 | Contrasting climatological surface cloud shortwave effect (SSCE) and its local sensitivity to warming ($\Delta \overline{SSCE}/\Delta \overline{T}$) between models with high and low $\Delta \overline{SSCE}/\Delta \overline{T}$. a, b** Spatial patterns of the climatological SSCE and $\Delta \overline{SSCE}/\Delta \overline{T}$ in the group of four models with the highest $\Delta \overline{SSCE}/\Delta \overline{T}$. **c, d,** Same as (**a, b**) but for the group of four models with the lowest $\Delta \overline{SSCE}/\Delta \overline{T}$. **e, f,** The differences in SSCE and $\Delta \overline{SSCE}/\Delta \overline{T}$ between the two groups.

the RCP8.5/SSP5-8.5 scenario from $6.34 \pm 3.53\%$ in the raw ensemble-model projection (black vertical error bar in Fig. 4c) to $7.03 \pm 2.59\%$ (red vertical error bar in Fig. 4c). This indicates a higher increase in global mean precipitation than the ensemble-model projection and reduces the intermodel uncertainty by 27%.

As the climatological SSCE affects $\Delta \overline{SSCE}/\Delta \overline{T}$ through its spatial gradient between warm and cold regions, we further construct a simpler index of the climatological SSCE pattern based on the difference in the mean SSCE between the region with SST>25°C and 0 < SST<25°C. This simple Index $D$ drops the need to compute the pattern correlation and can be easily applied to other warming scenarios and future CMIP ensembles. As shown in Fig. S9, Index $D$ effectively constrains $\Delta \overline{SSCE}/\Delta \overline{T}$ and $\Delta \bar{P}/\bar{P}$. The resultant ECs indicate similar results as those of Index C, demonstrating the robustness of the proposed EC.

### Relation and difference between different warming scenarios

Our EC also works well for the moderate-emission scenario of RCP4.5/SSP2-4.5. HS, CS and $\Delta \bar{P}/\bar{P}$ are all significantly correlated with $\Delta \overline{SSCE}/\Delta \overline{T}$ (Fig. S10), and $\Delta \bar{P}/\bar{P}$ can be constrained by the climatological index of SSCE (Fig. S11). The proposed EC constrains the 5%-95% uncertainty range of $\Delta \bar{P}/\bar{P}$ under RCP4.5/SSP2-4.5 from $4.19 \pm 2.28\%$ in the raw ensemble projection to $4.63 \pm 1.71\%$. This indicates a higher increase in global mean precipitation than the ensemble-model projection and reduces the projection uncertainty by 25%.

It is worthwhile to understand the subtle differences in the hydrological responses under RCP4.5/SSP2-4.5 and RCP8.5/SSP5-8.5 (Fig. S12,13,14). Different from the RCP8.5/SSP5-8.5 scenario which features a nearly linear $CO_2$ increase, the RCP4.5/SSP2-4.5 scenario stabilizes the $CO_2$ in the late 21st century. In this quasi-stabilized stage, the temperature-mediated effect (which increases precipitation over ocean) becomes more dominant compared to the effect of rapid adjustment (which reduces precipitation over ocean). As a result, unlike the nearly constant HS under RCP8.5/SSP5-8.5, HS under RCP4.5/SSP2-4.5 becomes larger in the late 21st century (Fig. S12a–d). The average HS is thus slightly higher under RCP4.5/SSP2-4.5. Nevertheless, HS values under these two warming scenarios are significantly correlated among models (Fig. S13). Furthermore, HS in the early and

late 21st century are highly correlated across models (Fig. S12e, f) and so is CS (Fig. S12g, h). For both periods, $\Delta \overline{SSCE}/\Delta \overline{T}$ is significantly correlated with HS, CS and $\Delta \bar{P}/\bar{P}$, and consequently, the climatological index of SSCE works well to constrain $\Delta \bar{P}/\bar{P}$ (Fig. S14).

The temporal evolutions of changes in different terms associated with surface energy balance are compared between RCP8.5/SSP5-8.5 and Abrupt4xCO2 (Fig. S15a–c). The response to Abrupt4xCO2 begins with rapid adjustment to the direct $CO_2$ radiative forcing and then experiences the temperature-mediated change (Fig. S15b). The response to RCP8.5/SSP5-8.5 shows the same trend as the temperature-mediated change in Abrupt4xCO2, indicating that the temperature-mediated effect dominates (Fig. S15c). For both RCP8.5/SSP5-8.5 and Abrupt4xCO2, the SSCE is the dominant factor that contributes to the uncertainty in the change of surface heat flux (Fig. S15d). As the uncertainty in rapid adjustment of SSCE is small, the uncertainty in $\Delta \overline{SSCE}/\Delta \overline{T}$ is dominated by the temperature-mediated feedback, i.e., SSCF (Fig. S15e, f). The temperature-mediated $\Delta \bar{P}/\bar{P}$ is significantly correlated with SSCF so the climatological Index of SSCE works well to constrain SSCF and consequently the temperature-mediated $\Delta \bar{P}/\bar{P}$ (Fig. S15g–i).

## Discussion

This study applies a surface-energy-balance perspective to constrain both the hydrological and climate sensitivities and subsequently constrain future precipitation increase under anthropogenic warming. We find that both the hydrological and climate sensitivities are significantly correlated with global surface cloud shortwave feedback, which can be further constrained using the observed pattern of climatological cloud effect. The resultant EC constrains the 5%-95% uncertainty range of $\Delta \bar{P}/\bar{P}$ under the high-emission (moderate-emission) scenario from $6.34 \pm 3.53\%$ ($4.19 \pm 2.28\%$) in the raw ensemble projection to $7.03 \pm 2.59\%$ ($4.63 \pm 1.71\%$). This indicates higher increases in global mean precipitation than the ensemble-model projection and reduces the projection uncertainty by 27% (25%). The constrained ranges of $\Delta \bar{P}/\bar{P}$ provide important information for impact assessments of climate change.

Our EC is different in several ways from the EC proposed by Ref. 22. First, we compare 1979–2005 and 2080–2100 while Ref. 22

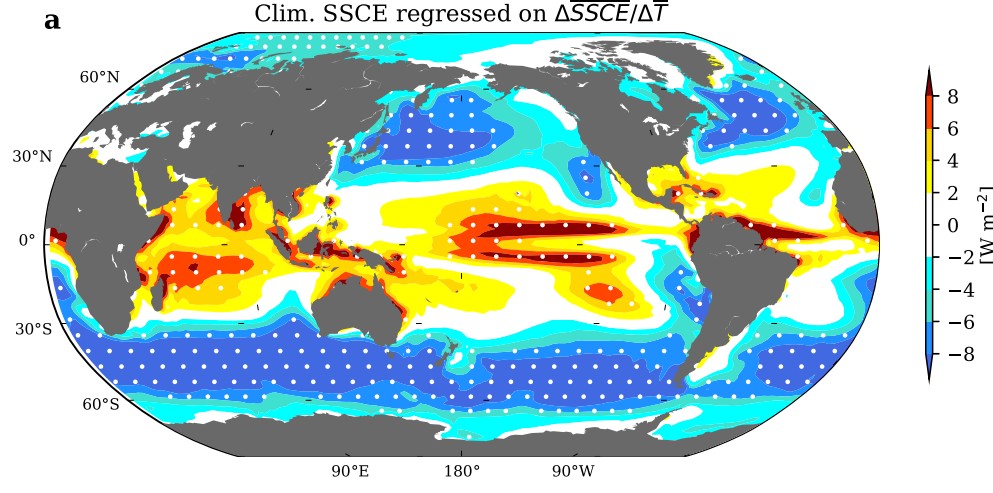

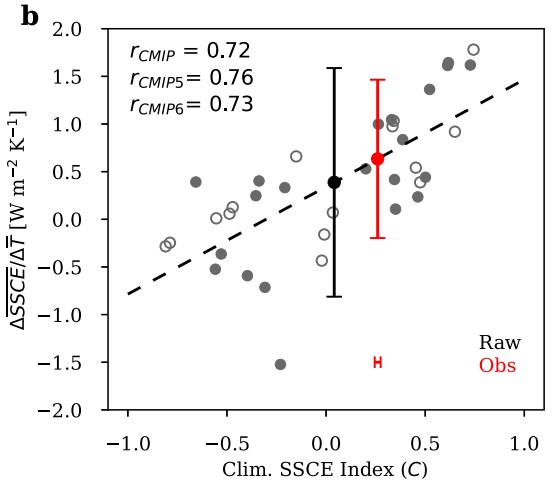

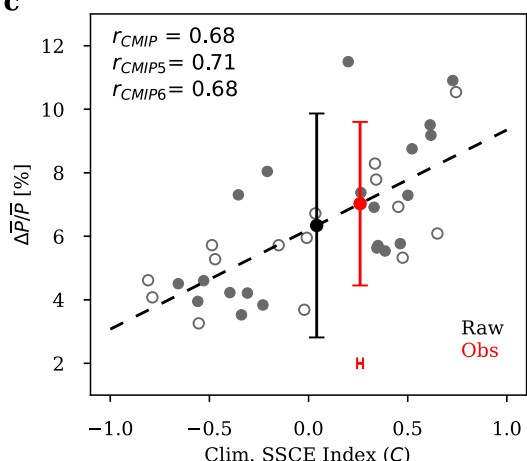

**Fig. 4 | An observational constraint on future global precipitation increase** $\Delta\bar{P}/\bar{P}$ **based on the climatological pattern of surface cloud shortwave effect (SSCE). a** The slope of the regression of the climatological SSCE onto global ocean SSCE change per unit warming ($\Delta\overline{SSCE}/\Delta\bar{T}$) across the CMIP models. White dots indicate where correlations are significant at $p = 0.05$. **b** Intermodel scatterplot between $C$ (the index of climatological SSCE pattern) and $\Delta\overline{SSCE}/\Delta\bar{T}$. **c** Intermodel scatterplot between $C$ and $\Delta\bar{P}/\bar{P}$. In (**b**, **c**), the open and filled dots are the CMIP5 and CMIP6 results, respectively. The horizontal error bar indicates the range of $C$ in the ISCCP and CERES satellite measurements. The vertical error bars indicate the 5%–95% range of the raw estimate (black; s.d. estimated from the model ensemble) and the emergent constraint (red; s.d. estimated from the unexplained variance after regression, Methods).

compares 1851-1900 and 2051-2100. We use 1979-2005 as it represents the climate we recently experienced, with concomitant satellite observations of radiative fluxes. Second, our EC utilizes both the HS and CS components to constrain $\Delta\bar{P}/\bar{P}$ while the EC in Ref. 22 mainly utilizes the CS component. Third, the final observable of our EC is a climatological pattern which by design presents less uncertainty than the estimate of forced temperature trend used by Ref. 22.. Due to the above contrasts, our EC reduces the uncertainty range of $\Delta\bar{P}/\bar{P}$ by ~25%, which is more than ~13% in Ref. 22. (Ref. 22. reports a ~ 25% reduction in variance, equivalent to a ~ 13% reduction in uncertainty range). Finally, our EC suggests a most-likely $\Delta\bar{P}/\bar{P}$ higher than the ensemble-model projection while their EC suggests lower. We achieve a higher $\Delta\bar{P}/\bar{P}$ as the observed pattern of SSCE indicates a higher $\Delta\overline{SSCE}/\Delta\bar{T}$ than the ensemble-model projection. Ref. 22. achieved a lower $\Delta\bar{P}/\bar{P}$ as the historical temperature trend in observations is lower than that simulated by models. It should be pointed out that future change under anthropogenic forcing should only be correlated with the forced component of the historical trend, which is not identical to the observed temperature trend due to natural variability. Specifically, as the Pacific Decadal Oscillation switched phase from positive to negative over 1979-2014, the

observed temperature trend is likely weaker than the forced trend[37] and accounting for this would increase the estimate of $\Delta\bar{P}/\bar{P}$ in Ref. 22. Caution is thus needed when interpreting the EC based solely on the observed temperature trend.

A key point of our work is studying CS and HS through the same surface-energy-balance perspective. Like the TOA perspective, the surface perspective links the uncertainty in CS to cloud shortwave effect. By linking precipitation to evaporation and interpreting evaporation through the Penman equation, the surface perspective attributes the uncertainty in HS to various factors including cloud shortwave effect. The surface perspective thus allows us to constrain CS and HS by the same observable and upon that derive an EC on $\Delta\bar{P}/\bar{P}$. In terms of understanding HS, the surface perspective highlights the contribution of the increasing energy allocation of surface latent heat flux to HS (Fig. 1d; Methods). This mechanism is not present in the conventional atmospheric-energy-balance perspective. While surface shortwave cloud effect is the dominant source of uncertainty in HS, other factors such as clear-sky shortwave radiation, longwave radiation and relative humidity also contribute (Fig. 1d and Fig. S4). Future studies will investigate details of their contributions and potential coupling.

## Methods

### Future projection of global climate models

Future changes under anthropogenic warming are projected by 36 climate models from the Coupled Model Intercomparison Project Phase 5 (CMIP5) and Phase 6 (CMIP6) under the Representative Concentration Pathway 4.5 and 8.5 warming scenarios (RCP4.5 and RCP8.5) for CMIP5 and under the Shared Socioeconomic Pathway 4.5 and 8.5 warming scenarios (SSP2-4.5 and SSP5-8.5) for CMIP6 (Table S1). Future changes are estimated from the differences in climatology between the historical period (1979-2005) and the future period (2080-2100). All outputs have been interpolated into a common $2.5^o \times 1.5^o$ grid before analyses.

### Effects of rapid adjustment and temperature-mediated feedback

The piControl and Abrupt4xCO2 simulations from the above 36 models are used to estimate the direct (i.e., rapid adjustment) and indirect (i.e., temperature-mediate) effect of the CO2 increase. Here, piControl refers to the pre-industrial control simulation with non-evolving pre-industrial conditions and it serves as a baseline for perturbation simulations that branch from it. Abrupt4xCO2 refers to the perturbation simulation in which an instantaneous quadrupling of CO2 is imposed and then held fixed.

The temperature-mediated feedback is estimated from the change normalized by ocean warming after year 1, i.e.,

$$\text{Temp} - \text{mediated Feedback} = \frac{\triangle z_{1-80}}{\triangle T_{1-80}} \quad (1)$$

where $\triangle T_{1-80}$ is the increase in global mean ocean surface temperature from year 1 to year 80 and $\triangle z_{1-80}$ is the change in global mean of a specific variable (i.e., SSCE or precipitation) from year 1 to year 80.

The effect of rapid adjustment is computed as

$$\text{Rapid Adjustment} = \triangle z_1 - \text{Feedback} \times \triangle T_1 \quad (2)$$

where $\triangle z_1$ is the difference between year 1 in Abrupt4xCO2 and the piControl mean, and Feedback $\times \triangle T_1$ represents the temperature-mediated effect in year 1, estimated as the temperature-mediated feedback multiplied by the temperature increase in year 1. When comparing the relative importance of rapid adjustment and the temperature-mediated feedback for the RCP/SSP scenario, we normalize the effect of rapid adjustment by the total degree of ocean warming.

### The reformulated Penman equation

The Penmen equation estimates evaporation ($E$) over open water as

$$L_v E = \frac{\triangle}{\triangle + \gamma}(SW + LW - G) + \frac{\gamma}{\triangle + \gamma}\kappa e^*(1 - RH) \quad (3)$$

where $L_v$ is the latent heat of vaporization, $e^*$ is the near-surface saturation vapor pressure, $\triangle \equiv \frac{\partial e^*}{\partial T} = \frac{L_v e^*}{R_v T^2}$ measures the increase of $e^*$ with temperature, $\gamma$ is the psychometric constant, $\kappa \equiv \frac{\rho C_p}{\gamma r_a}$, $r_a$ is the aerodynamic resistance, $T$ and $RH$ are the near-surface air temperature and relative humidity, $SW$ and $LW$ are the net downward surface shortwave and longwave radiative fluxes, and $G$ is the ocean heat uptake. Following Refs. 25,29, the Penman evaporation can be rewritten as,

$$L_v E = f[SW + LW - G + \xi(1 - RH)] \quad (4)$$

where $f \equiv \frac{\triangle}{\triangle + \gamma}$ is the energy allocation factor and $\xi = \frac{R_v T^2}{L_v}k\gamma$ scales the $RH$ effect.

As saturation vapor pressure $e^*$ increases exponentially with warming, $\triangle$ increases with warming at the rate of $\frac{dln\triangle}{dT} = \frac{L_v}{R_v T^2} - \frac{2}{T} \cong 5.8\%$ $K^{-1}$, and consequently the energy allocation factor $f \equiv \frac{\triangle}{\triangle + \gamma}$ increases with warmer temperature at the rate of $\frac{dlnf}{dT} \cong (1-f)\frac{dln\triangle}{dT}$. That is, a warmer temperature will increase the energy allocation of total available energy to surface latent flux and contribute to future evaporation increase.

The Penman-based estimates of evaporation are computed from Eq. 4 using monthly mean air temperature, relative humidity, surface radiative flux, ocean heat uptake and surface wind speed in each model. The aerodynamic resistance in $\kappa$ is computed from the surface wind speed as $r_a = 1. \times 10^{-3}(1 + 0.7U)$, which fits the simulation climatology well. Because it is difficult to directly compute the ocean heat uptake, $G$ is diagnosed as the residual of the surface energy balance, i.e., $G = SW + LW - L_v E - SH$. In climatology of models (1979-2005), the global mean G is ~1.34±0.8 W m$^{-2}$ while $SW + LW$ is ~115±3.5 W m$^{-2}$ and $SSCE$ is 59±5.5 W m$^{-2}$. When estimating the contribution of a particular factor to the ocean evaporation change, we change its values in the Penman equation from those in historical climate to those in future climate while other factors are unchanged.

### Estimation of uncertainty ranges

For the raw multi-model ensemble estimate, the 5%-95% uncertainty range is estimated from the $\pm 1.64$ s.d. of the intermodel spread, assuming the spread generally follows a Gaussian distribution. For the emergent constraint, the uncertainty range is estimated using the s.d. of residual variance after regression and we compute the s.d. of residual variance using the hierarchical statistical framework following Refs. 22,38. Let z be the future changes, x be the parameter $C$ in climate models, and y be the observed values of $C$ (here we only have two data points from CERES and ISCCP). The observationally constrained mean $E(z,|,y)$ and s.d. $\delta(z,|,y)$ is calculated by the following equations,

$$E(z|y) = u_z + \frac{\rho \delta_x \delta_z}{\delta_x^2 + \delta_y^2}\left(u_y - u_x\right) \quad (5)$$

$$\delta(z|y) = \delta_z \sqrt{1 - \frac{\rho^2}{1 + \delta_y^2/\delta_x^2}} \quad (6)$$

where $u$ and $\delta$ refer to the mean and s.d., and $\rho$ is the correlation between x and z.

### Reporting summary

Further information on research design is available in the Nature Portfolio Reporting Summary linked to this article.

## Data availability

The CMIP5 and CMIP6 outputs are available from the Earth System Grid Federation (ESGF) Portal at https://esgf-node.llnl.gov/projects. The CERES dataset is available at https://ceres.larc.nasa.gov/data/. The ISCCP dataset is available at https://isccp.giss.nasa.gov/projects/flux.html. The raw data underlying the figures are available at https://doi.org/10.5281/zenodo.8383970.

## Code availability

The scripts for analyses and generating figures are available at https://doi.org/10.5281/zenodo.8383970.

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

## Acknowledgements

We acknowledge the WCRP Working Group on Coupled Modeling, which is responsible for CMIP, and the climate modeling groups for producing and making available their model outputs. This study was supported by Office of Science, U.S. Department of Energy Biological and Environmental Research as part of the Regional and Global Model Analysis program area. The Pacific Northwest National Laboratory (PNNL) is operated for DOE by Battelle Memorial Institute under contract DE-AC05-76RLO1830. N.S. was supported by NSF Grant AGS–1954663. This research used resources of the National Energy Research Scientific Computing Center, which is supported by the Office of Science of the U.S. Department of Energy under Contract No. DE-AC02-05CH1123.

## Author contributions

W.Z. designed the research and conducted the analysis. R.L., N.S. and J.L. contributed to improving the analysis and interpretation. W.Z. wrote the first draft and all the authors edited the paper.

## Competing interests

The authors declare no competing interests.
