## [Peer Review File · Nature Communications]

REVIEWER COMMENTS

Reviewer #1 (Remarks to the Author):

Summary:

This paper aims to constrain the projection uncertainty of global mean precipitation change, by constraining its two components – the hydrological sensitivity ($\Delta P/\Delta T$) and the climate sensitivity. Using a unified surface-energy-balance framework, it is found that both sensitivities are significantly correlated with surface cloud shortwave feedback, which can be constrained using the observed pattern of climatological cloud effect. This is a physically strong emergent constraint. It reduces the projection uncertainty by more than 25% in the CMIP5/6 ensemble.

The emergent constraint is effective and novel, because it applies to both the hydrological sensitivity and the climate sensitivity. The main conclusions are well supported by the analyses. My major comments focus on the analysis of hydrological sensitivity, including the applicability of the emergent constraint in transient and “quasi-stabilized” climate change scenarios, and the scenario dependency of hydrological sensitivity. These are fundamental issues in the hydrological cycle change.

Major comments:

1. The emergent constraint of hydrological sensitivity ($\Delta P/\Delta T$) is applied to the median emission scenario (RCP4.5/SSP2-4.5). Under RCP4.5/SSP2-4.5 scenarios, global mean temperature increases in the near-term and then stabilizes. That is, it includes two types of climate change under RCP4.5/SSP2-4.5 – transient climate change in the first stage and “quasi-stabilized” climate change in the second stage (I use “quasi-stabilized” because only the global mean temperature is stabilized but not in the ocean state). Thus, the hydrological sensitivity ($\Delta P/\Delta T$) is different in these two stages. How does this affect the constraint? Does the emergent constraint apply to both stages?
2. As the hydrological sensitivity ($\Delta P/\Delta T$) depends on external forcings, it should differ between different emission scenarios. Please add some discussions on the scenario dependency of hydrological sensitivity (in raw simulations and in constrained results).
3. There have been several studies providing constrained projections of hydrological sensitivity or global mean precipitation change. How do the quantitative results compare with each other? Please add some discussions on this.

Minor comments:

4. Fig. S1: Please clarify which experiments were used for CMIP5 and CMIP6.

5. L126 and L157: Please use the full term “global surface shortwave cloud feedback” instead of GSSCF in the subtitle.

Reviewer #2 (Remarks to the Author):

This paper presents a new emergent constraint on global hydrologic sensitivity by utilizing the relationship between the pattern of surface climatological shortwave cloud radiative effect and the projected global precipitation response under future emission scenarios. The relationship is rooted in the physics that tie the climatological cloud pattern, shortwave cloud feedback, oceanic energy budget (and thus global evaporation), and climate sensitivity together. To my knowledge, the relationship between global precipitation and shortwave cloud effect has not been discovered/shown previously. An emergent constraint utilizing satellite measurements of the climatological shortwave cloud effect pattern results in a small constraint on the model spread of the projected global precipitation response. While the final constraint is marginal, I think this study is valuable for revealing a new emergent constraint that can be utilized in other model ensembles and observational datasets in the future. Also, the study uses a large and robust collection of climate models and reasonable metrics to perform the constraint. Nonetheless, I think some revisions are warranted to improve the robustness and physical interpretation of the results, as described below. I think the paper may be suitable for publication in Nature Communications if the authors can address these concerns.

1. A reasonable assumption may be that the relationships shown for the RCP emissions scenarios here (e.g., Figs. 2, 4, S6, S7) are rooted in the response of the relevant quantities (clouds, precipitation, evaporation) to planetary warming. In other words, I wonder if the temperature-mediated responses [i.e., the changes that only result from global (predominantly oceanic) warming and that exclude potential contributions from rapid adjustments due to changes in the forcing itself (from greenhouse gases, aerosols, etc.)] are largely responsible for the relationships. Is this true? This could be diagnosed with simulations that abruptly increase CO₂ initially then leave it constant, or with simulations that just increase SST but not radiatively active atmospheric constituents, allowing one to examine the response to just the warming itself (e.g., see refs. 17, 18, and 21). Showing that these inter-model relationships also exist in the temperature-mediated framework would help to 1) show their robustness, and 2) better pin down their physical basis. I suggest the authors consider doing this (and perhaps showing the results as supplementary figures), which would help strengthen their conclusions and allow for a more direct comparison with previous studies that constrain hydrologic sensitivity on the basis of temperature-mediated responses.

2. Regarding the evaporation calculations: can the authors expand on the ocean heat uptake (G) calculation a bit more in methods? If it is computed as a residual, this implies that it uses model-generated SW, LW, and E. How then is model-generated E different from the E computed from the Penman equation? Is one expected to be a better estimate than the other? Also, do the SW and LW terms appear twice or cancel out if one were to expand G in equation (2). In short, I wonder how strongly the model-generated and Penman E are dependent on each other (due to the way G is computed) and whether this dependence could explain the apparently good agreement between these quantities (e.g., Fig. S2).

3. Has the sensitivity to the definition of C (the metric used for the emergent constraint) been explored? Currently, C depends on the relationship between GSSCF and climatological SSCE itself (Fig. 4a). Would a metric that is independent of this relationship (say, just the difference between climatological SSCE between the low and high latitudes) give the same tight relationships as in (b) and (c) of Fig. 4? Not only would this demonstrate robustness, but also allow one to compute the metric in other ensembles without the need to first generate Fig. 4a based on that ensemble.

Specific/minor comments:

L45-46: “a more recent analysis suggests that the statistical relationships underlying these two ECs do not hold up among the latest generation of climate models” – This claim is not entirely true. As stated in the abstract of ref. 21: “The constraint based on clear-sky shortwave absorption sensitivity to water vapor has weakened, and it is argued that a proposed constraint based on surface low cloud longwave radiative effects does not apply to HS.” Ref. 21 does not suggest that both ECs no longer apply or that all key related relationships no longer exist.

L52-54: “The rationale is that future precipitation increase is correlated with future temperature increase and the latter is correlated with the forced trend over recent decades” – Did you mean to say “the former” rather than “the latter” is correlated...?

L54-55: “That said, only the CS part has been utilized in the EC to constrain DP/P while the potential to constrain the HS part has not been harnessed.” I'm not sure this is true since that study (ref. 22), according to its abstract, uses trends in DP itself to constrain future DP. It is thus constraining the total DP directly, which has contributions from HS and CS as the authors state earlier.

Figs. 1, S2, and possible others: It would help to state the time periods that are plotted in the figure captions.

L126: I would avoid using such a long acronym (GSSCF) in the section title, especially since this acronym has not been defined yet.

L158-160: "The shortwave cloud feedback, widely considered as a major uncertainty source for CS, has been often studied from the perspective of the top of the atmosphere (TOA)." What is the relationship between the TOA and SFC SW cloud feedback? It is assumed here that these should be essentially the same, qualitatively and perhaps even quantitatively, to some extent? If so, this should be stated explicitly.

Reviewer #3 (Remarks to the Author):

Several major revisions and comments are provided for the manuscript, as I found it unconvincing:

Abstract: what exactly is meant by climatological cloud effect? This is very vague and I don't know what exactly it is about clouds that is meant

Line 27: Give a clear definition of what $\Delta P/P$ stands for (the fractional change in mean precip normalized by the mean precip)

Line 33: "varying by a factor of 2" is unclear language.

Line 41: Emergent constraints should be constructed only with observable quantities, but this sentence seems to imply anything can be used?

Line 48-49: "under different frameworks" and "different but likely dependent observables" are oddly worded and unclear.

Line 51: Not sure this is necessary, seems like a rehash of previous paragraph? Also co-dependencies of terms within the EC as explained in previous paragraph?

Line 59: All ECs should be physically guided, so this statement is unnecessary. That said, why a surface energy balance framework – what is new and of value here, as opposed to ECs based on the TOA budget, for example.

Lines 36-57: The manuscript could be better motivated. It is good to explain what ECs are, but the focus is supposed to be on finding an EC for precipitation increases with warming, so why spend so much time discussing ECs for CS and HS? It's distracting, focus instead on ECs for precipitation increase specifically. That said, I believe there has been other rather recent work in addition to Ref. 22 on ECs for precipitation increase, and the manuscript alludes to how this EC is different from other studies; this should be further discussed. What makes this framework for developing an EC in this manuscript new and of value? The motivation for using ocean evaporation and the surface energy budget is particularly missing.

Line 60-62: It's not necessary for an EC to constrain both HS and CS first in order to well constrain future precipitation increases (so long as a physical relationship exists between the observed quantity and the precipitation increase). This seems convoluted and potentially misguided. Why this approach?

Line 124: compensation from other factors maybe deserves to be explored – if we reduced uncertainty in SSCE then how much would it really matter given these other factors (ok, at least the discussion section acknowledges this, but this issue still needs to be addressed).

Line 131: Calling the change in SSCE with warming a “cloud feedback” is misleading, since this includes feedbacks and adjustments but isn't quite itself a feedback, it's a change in shortwave cloud radiative effects as seen by the surface. A different analysis is needed to isolate the surface shortwave cloud feedbacks.

Lines 145: There actually seems to a deterioration between the CMIP5 and CMIP6 ensembles in the correlation between GSSCF and change in evaporation and change in precipitation with warming, but especially for precipitation. The deterioration is fairly large for precipitation, and appears to be driven by two models. Which are these and why do they depart so starkly from the other CMIP6 models? Doesn't this negatively impact the EC constructed for precipitation, or is $r=0.67$ still a high enough correlation?

Line 158: This approach seems odd – since GSSCF is correlated with $\Delta P/P$, why not just find an observable variable that is highly correlated or linked to shortwave cloud feedback and build the EC for precipitation with that variable directly? That seems to me to be the standard way to construct an EC, rather than build an EC on top of an EC (constraining GSSCF to then constrain precipitation change), which may be a misguided approach. If this is what was done, this needs to be made more clear, because it does not read this way.

Line 160: Need to make clear what understanding/advancement is gained by examining from a surface energy balance perspective rather than TOA.

Line 162: Again, what exactly is meant by climatological cloud pattern? Just their geographical distribution, and/or optical thickness, etc.?

Lines 167-168: Why ignoring the models in between these two groups?

Line 224: does EC break down for low emission scenario and why?

How does this compare to ECs based on TOA framework?? Like in ref 22

Line 237: this wasn't shown for f, wasn't it the definition?? Also this wasn't shown or discussed at all really anyway

Fig. 4, remainder of EC construction, and Discussion: the EC presented in Fig. 4 seems veeeeeeery hand-wavy. The climatological SSCE index is heavily derived parameter, based on quantities that are observed but can have a high uncertainty, while ECs ideally should be constructed with observable parameters/variables that at the least are not so derived or directly measured, and the observable's uncertainty range tells something about the plausible range of the parameter being constrained by highlighting its values that fall into that range. I do not see that with the EC as presented in Fig. 4 for precipitation change with warming. The horizontal error bars in Fig. 4 appear to be missing in fact, so that this plausible range of precipitation change values does not seem like it can be easily estimated despite the uncertainty value given in the text. It's really the horizontal error bars on the climatological index and NOT the vertical bars that we're after with an EC anyway, and so the uncertainty range given in the text appears to be misestimated; we want to see which values of the precipitation change are associated with the range of index values over its uncertainty range – this is the uncertainty range on the precipitation change rather than the calculation underlying the vertical error bars in Fig. 4. I remain unconvinced by this EC because of the amount of derivation required and the misguided uncertainty/constraint estimation.

Minor Comments:

Fig. S4: what is the x-axis?? The model “number?”

Dear Reviewers:

Thank you so much for your insightful comments and detailed suggestions. They are very helpful to us for improving the manuscript. We have revised the manuscript accordingly.

The major changes include:

1. Diagnose the effects of rapid adjustment and the temperature-mediated feedback based on the piControl and Abrupt4xCO2 simulations of CMIP5/6 models, and use them to further understand the response under the RCP scenarios
2. Illustrate the relation and difference among different warming scenarios and periods, and demonstrate the usefulness of the proposed EC to different scenarios and periods.
3. Introduce a simple index of the climatological pattern of SSCE based on the SSCE difference between regions with $SST > 25^{\circ}\text{C}$ and $0^{\circ}\text{C} < SST < 25^{\circ}\text{C}$. This simple Index D does not involve the computation of pattern correlation and shows the robustness of the EC.
4. Add more detailed comparison with previous ECs on HS and $\Delta\bar{P}/\bar{P}$ and highlight the advantage of the proposed EC
5. The uncertainty of the observations is already included in Fig. 4 (for Index C) and Fig. S9 (for the new simple Index D). As a benefit of using the climatological pattern as the final observable, the uncertainty from the observations is very small so the uncertainty of the EC is mainly from the unexplained variance by the correlation.
6. Correct the computation of Index C by only using the ocean grid points (as we are talking about surface energy balance and evaporation over ocean). This correction slightly increases the most-likely $\Delta\bar{P}/\bar{P}$ constrained by the EC.

Our point-by-point responses are provided below (the original review in black and the authors' responses in red).

Response to Reviewer #1:

This paper aims to constrain the projection uncertainty of global mean precipitation change, by constraining its two components – the hydrological sensitivity ($\Delta P/\Delta T$) and the climate sensitivity. Using a unified surface-energy-balance framework, it is found that both sensitivities are significantly correlated with surface cloud shortwave feedback, which can be constrained using the observed pattern of climatological cloud effect. This is a physically strong emergent constraint. It reduces the projection uncertainty by more than 25% in the CMIP5/6 ensemble.

The emergent constraint is effective and novel, because it applies to both the hydrological sensitivity and the climate sensitivity. The main conclusions are well supported by the analyses. My major comments focus on the analysis of hydrological sensitivity, including the applicability of the emergent constraint in transient and “quasi-stabilized” climate change scenarios, and the scenario dependency of hydrological sensitivity. These are fundamental issues in the hydrological cycle change.

Major comments:

1. The emergent constraint of hydrological sensitivity ($\Delta P/\Delta T$) is applied to the median emission scenario (RCP4.5/SSP2-4.5). Under RCP4.5/SSP2-4.5 scenarios, global mean temperature increases in the near-term and then stabilizes. That is, it includes two types of climate change under RCP4.5/SSP2-4.5 – transient climate change in the first stage and “quasi-stabilized” climate change in the second stage (I use “quasi-stabilized” because only the global mean temperature is stabilized but not in the ocean state). Thus, the hydrological sensitivity ($\Delta P/\Delta T$) is different in these two stages. How does this affect the constraint? Does the emergent constraint apply to both stages?

We thank the reviewer for raising this important question. In response to your comment, we have conducted additional analyses and added supplementary figures to illustrate the two stages of climate change in RCP4.5 and demonstrate that the EC applies to both stages.

The climate response to the RCP forcing can be understood as a combination of the direct (i.e., rapid adjustment) and indirect (i.e., temperature-mediated) effects. Fig. R1a shows the temporal evolutions of ΔT and $\Delta P/P$ in RCP4.5, RCP8.5 and Abrupt4xCO₂ and Fig. R1b plots ΔT versus $\Delta P/P$ over time (the slope of the regression line indicates HS). According to Abrupt4xCO₂ in Fig. R1a, the direct effect reduces P (as seen in the rapid response at year 1) while the temperature-mediated effect increases P (as seen in the response after year 1). For RCP4.5, the CO₂ increase stabilizes in the late 21st century, so the temperature-mediated effect becomes more dominant than the effect of rapid adjustment, and HS becomes larger (Fig. R1b). Nevertheless, HS in the early (2049-2069 minus 1979-2005) and late (2080-2100 minus 2049-2069) periods are highly correlated among models (Fig. R1e,f) and so are CS in the two periods (Fig. R1g,h). As shown in Fig. R2, for both stages, the intermodel uncertainty in both HS and CS are significantly correlated with that of $\overline{\Delta SSCE}/\overline{\Delta T}$, and Index C works well to constrain $\Delta P/P$.

We have added a new section to discuss the relation and difference among different warming scenarios. The early and late stages in RCP4.5 are discussed in the second paragraph of that section. We have included Fig. R1 and Fig. R2 as Fig. S12 and Fig. S14 in Supplementary Information.

Figure R1: a, Temporal evolution of the global mean temperature ($\Delta\bar{T}$, solid) and precipitation ($\Delta\bar{P}/\bar{P}$, dashed) under RCP8.5 (red), RCP4.5 (blue) and Abrupt4xCO2 (black). **b**, Relationship between changes in temperature and precipitation (the slope indicates hydrological sensitivity $\Delta\bar{P}/\bar{P}/\Delta\bar{T}$) under RCP8.5 (red), RCP4.5 (blue) and Abrupt4xCO2 (black). **c**, Relationship between changes in temperature and precipitation under RCP8.5 for individual models. The slopes of HS=1.2% K-1 and HS=3.2% K-1 are plotted for reference. **d**. As c but for RCP4.5. **e**. Intermodel scatterplot of HS in the early (Mid-Future minus Historical, i.e., 2049-2069 minus 1979-2005, denoted as MF - H) and late (Future minus Mid-Future, i.e., 2080-2100 minus 2049-2069, denoted as F - MF) periods of RCP8.5. **f**. As e but for RCP4.5. **g,h**, As e,f but for CS. The open and filled dots are the CMIP5 and CMIP6 results, respectively.

Figure R2: a,e, Intermodel scatterplot between $\Delta\overline{SSCE}/\Delta\bar{T}$ and $\Delta\bar{E}_o/\bar{E}_o/\Delta\bar{T}$. **b,f**, Intermodel scatterplot between $\Delta\overline{SSCE}/\Delta\bar{T}$ and $\Delta\bar{T}$. **c,g**, Intermodel scatterplot between $\Delta\overline{SSCE}/\Delta\bar{T}$ and $\Delta\bar{P}/\bar{P}$. **d,h**, Intermodel scatterplot between Index C and $\Delta\bar{P}/\bar{P}$. **a-d** are for the early period of the 21st century under RCP4.5 (Mid-Future minus Historical, 2049-2069 minus 1979-2005) while **e-h** are for the late period of the 21st century under RCP4.5 (Future minus Mid-Future, 2080-2100 minus 2049-2069). The open and filled dots are the CMIP5 and CMIP6 results, respectively.

2. As the hydrological sensitivity ($\Delta P/\Delta T$) depends on external forcings, it should differ between different emission scenarios. Please add some discussions on the scenario dependency of hydrological sensitivity (in raw simulations and in constrained results).

As the CO₂ increase stabilizes in the late stage of RCP4.5, the temperature-mediated effect becomes more dominant, and HS becomes larger than that in RCP8.5 which features a nearly linear CO₂ increase. Nevertheless, HS is highly correlated between RCP4.5 and RCP8.5 (Fig. R3). We have added a new section “Relation and difference among different warming scenarios” to address Comments 1 and 2. The application of the EC to RCP4.5 is described in the first paragraph of that section and summarized in Fig. S11 and Fig. S11.

Figure. R3: Intermodel scatterplot between HS's in RCP8.5 and RCP4.5

In response to both Comments 1 and 2, we would like to note that while the temporal evolution of the anthropogenic forcing differs among different periods and warming scenarios, all the models run share the same forcing given a specific scenario/period and here we build ECs to constrain the intermodel uncertainty. The ECs work well for different periods and warming scenarios, because the intermodel uncertainty in $\Delta \bar{P}/\bar{P}$ is consistently dominated by the uncertainty in $\Delta \overline{SSCE}/\Delta \bar{T}$, and $\Delta \overline{SSCE}/\Delta \bar{T}$ is correlated with the climatological pattern of SSCE.

3. There have been several studies providing constrained projections of hydrological sensitivity or global mean precipitation change. How do the quantitative results compare with each other? Please add some discussions on this.

In the manuscript, we have reviewed two constraints on HS (Ref. 17 and 18) and one constraint on global mean precipitation increase (Ref. 22). These three references, to our knowledge, present the most updated ECs on future precipitation increase.

Ref. 17 and 18 are based on CMIP5. Ref. 21 shows that, in CMIP6, the correlation proposed by Ref. 17 has substantially weakened ($r = -.045$) and the correlation proposed by Ref. 18 no longer holds ($r = -0.01$). We have now explicitly mentioned these correlations in Introduction. We have discussed in detail how our EC is different from the EC in Ref. 22 (second paragraph in Discussion). Because our EC constrains both HS and CS and because our final observable is a climatological pattern instead of a trend which is more uncertain, our EC reduces ~25% of the intermodel uncertainty in $\Delta P/P$, while Ref. 22 reduces only ~13%. Note that Ref. 22 reports a ~25% “reduction in variance”, which is equivalent to a ~13% reduction in uncertainty range.

Minor comments:

4. Fig. S1: Please clarify which experiments were used for CMIP5 and CMIP6.

We have clarified that Fig. S1 is based on RCP8.5. Also, we have stated in the last paragraph of Introduction that the main result focuses on RCP8.5. The results of RCP4.5 are discussed in the section before Discussion.

5. L126 and L157: Please use the full term “global surface shortwave cloud feedback” instead of GSSCF in the subtitle.

We have modified the two subtitles to “Surface shortwave cloud feedback regulates future precipitation changes by affecting both CS and HS” and “An emergent constraint on future precipitation increase”.

Response to Reviewer #2:

This paper presents a new emergent constraint on global hydrologic sensitivity by utilizing the relationship between the pattern of surface climatological shortwave cloud radiative effect and the projected global precipitation response under future emission scenarios. The relationship is rooted in the physics that tie the climatological cloud pattern, shortwave cloud feedback, oceanic energy budget (and thus global evaporation), and climate sensitivity together. To my knowledge, the relationship between global precipitation and shortwave cloud effect has not been discovered/shown previously. An emergent constraint utilizing satellite measurements of the climatological shortwave cloud effect pattern results in a small constraint on the model spread of the projected global precipitation response. While the final constraint is marginal, I think this study is valuable for revealing a new emergent constraint that can be utilized in other model ensembles and observational datasets in the future. Also, the study uses a large and robust collection of climate models and reasonable metrics to perform the constraint. Nonetheless, I think some revisions are warranted to improve the robustness and physical interpretation of the results, as described below. I think the paper may be suitable for publication in Nature Communications if the authors can address these concerns.

Thank you for your review and appreciation of our work. By constraining both CS and HS, our ECs reduce the intermodel uncertainty in $\Delta P/P$ by 25%-27%, which to our knowledge is more effective than any previously proposed ECs. Also, our EC indicates a higher most-likely $\Delta P/P$ than the mean ensemble-model projection. These results would provide important information for credible assessment of hydrological impacts of climate change.

1. A reasonable assumption may be that the relationships shown for the RCP emissions scenarios here (e.g., Figs. 2, 4, S6, S7) are rooted in the response of the relevant quantities (clouds, precipitation, evaporation) to planetary warming. In other words, I wonder if the temperature-mediated responses [i.e., the changes that only result from global (predominantly oceanic) warming and that exclude potential contributions from rapid adjustments due to changes in the forcing itself (from greenhouse gases, aerosols, etc.)] are largely responsible for the relationships. Is this true? This could be diagnosed

with simulations that abruptly increase CO2 initially then leave it constant, or with simulations that just increase SST but not radiatively active atmospheric constituents, allowing one to examine the response to just the warming itself (e.g., see refs. 17, 18, and 21). Showing that these inter-model relationships also exist in the temperature-mediated framework would help to 1) show their robustness, and 2) better pin down their physical basis. I suggest the authors consider doing this (and perhaps showing the results as supplementary figures), which would help strengthen their conclusions and allow for a more direct comparison with previous studies that constrain hydrologic sensitivity on the basis of temperature-mediated responses.

Thank you so much for your suggestion! We have analyzed the Abrupt4xCO2 simulation and related it to the RCP scenarios. We find that the temperature-mediated part is largely responsible for the proposed constraint under RCP, and the constraint does apply to the temperature-mediated change in Abrupt4xCO2.

We use the piControl and Abrupt4xCO2 simulations to estimate the direct (i.e., rapid adjustment) and indirect (i.e., temperature-mediated) effect of the CO2 increase. The temperature-mediated feedback is estimated from the change normalized by ocean warming after year 1, i.e.,

$$\text{Feedback} = \frac{\Delta z_{1-80}}{\Delta T_{1-80}}$$

where ΔT_{1-80} is the increase in global mean ocean surface temperature from year 1 to year 80 and Δz_{1-80} is the change in global mean of a specific variable (i.e., SSCE or precipitation) from year 1 to year 80. The effect of rapid adjustment is computed as

$$\text{Rapid Adjustment} = \Delta z_1 - \text{Feedback} \times \Delta T_1$$

where Δz_1 is the difference between year 1 in Abrupt4xCO2 and the piControl mean, and $\text{Feedback} \times \Delta T_1$ represents the temperature-mediated effect in year 1, estimated as the temperature-mediated feedback multiplied by the temperature increase in year 1. When comparing the relative importance of rapid adjustment and temperature-mediated feedback, we normalize the effect of rapid adjustment by the total degree of ocean warming.

Fig. R4a depicts the different CO2 forcing under Abrupt4xCO2 and RCP8.5, and Fig. R4b,c show the temporal evolution of changes in different terms associated with surface energy balance under Abrupt4xCO2 and RCP8.5, respectively. The Abrupt4xCO2 scenario begins with rapid adjustment and then experiences the temperature-mediated changes. The response of individual terms under RCP8.5 shows the same trend as the temperature-mediated change in Abrupt4xCO2, indicating the temperature-mediated effect dominates over the direct CO2 effect. The intermodel s.d. of the changes in these terms is summarized in Fig. R4d, for the rapid adjustment and temperature-mediated part of Abrupt4xCO2 and for RCP8.5. We can see that SSCE is the dominating factor that contributes to the uncertainty in the change of HFS (latent + sensible heat flux). Fig. R4e compares the rapid adjustment and the temperature-mediated feedback of SSCE in individual models. The effect of rapid adjustment is much less uncertain compared to the temperature-mediated feedback (i.e., SSCF). As a result, $\Delta\overline{SSCE}/\Delta\bar{T}$ in RCP8.5 is highly correlated with SSCF diagnosed from Abrupt4xCO2 among models (Fig. R4f, $r=0.9$). Subsequently, $\Delta\bar{E}_o/\bar{E}_o/\Delta\bar{T}$, $\Delta\bar{T}$, $\Delta\bar{E}_o/\bar{E}_o$ and $\Delta\bar{P}/\bar{P}$ in RCP8.5 are all significantly correlated with SSCF (Fig. R5). The temperature-mediated $\Delta\bar{P}/\bar{P}$ in Abrupt4xCO2 is significantly correlated with SSCF and the climatological Index of SSCE works well to constrain SSCF and consequently the temperature-mediated $\Delta\bar{P}/\bar{P}$ (Fig. R4g-i).

We have added a new paragraph (Line 166-173, copied below) to highlight that the temperature-mediated feedback dominates the uncertainty in the change of SSCE. In this paragraph, Fig. R4e,f is included as Fig. 3e,f and Fig. R5 is included as Fig. S7.

“ $\Delta\overline{SSCE}/\Delta\bar{T}$ is the change in global ocean SSCE per unit warming under the RCP scenario, which includes the effects of both rapid adjustment and temperature-mediated feedback. By diagnosing these two effects from the piControl and Abrupt4xCO2 experiments (Methods), we can see that the effect of rapid adjustment of SSCE (normalized by ocean warming) is much less uncertain compared to the temperature-mediated feedback, i.e., surface shortwave cloud feedback or SSCF (Fig. 2e). As a result, the intermodel uncertainty in $\Delta\overline{SSCE}/\Delta\bar{T}$ is largely due to that of SSCF, with the intermodel correlation

between $\overline{\Delta SSC E} / \Delta \bar{T}$ and SSCF at $r = 0.9$ (Fig. 2f). Subsequently, $\Delta \bar{E}_o / \bar{E}_o / \Delta \bar{T}$, $\Delta \bar{T}$, $\Delta \bar{E}_o / \bar{E}_o$ and $\Delta \bar{P} / \bar{P}$ are all significantly correlated with SSCF (Fig. S7).”

We have also added a new section before Discussion to talk about the relation and difference among different warming scenarios, with the help of this decomposition of direct and indirect effects. Fig. R4 is included as Fig. S15 in this section.

Figure R4: a, Temporal evolutions of the atmospheric CO2 concentration under Abrupt4xCO2 and

RCP8.5. **b**, Temporal evolutions of different terms associated with surface energy balance, i.e., global mean SW_{clr} , SW_{ctr} , LW , G and HFS , to Abrupt4xCO₂. The solid line shows the ensemble model mean and the shading indicates the intermodel s.d. **c**, As b but for RCP8.5. **d**, Ensemble model mean and intermodel s.d. of the changes in SW_{clr} , SW_{ctr} , LW , G and HFS under Abrupt4xCO₂ (separated into rapid adjustment and temperature-mediated effects) and under RCP8.5. **e**, Intermodel scatterplot between rapid adjustment and temperature-mediated feedback of SSCE under Abrupt4xCO₂. To compare their relative effects, the effect of rapid adjustment has been normalized by global ocean warming. **f**, Intermodel scatterplot between the temperature-mediated feedback of SSCE diagnosed from Abrupt4xCO₂ (i.e., SSCF) and $\Delta\overline{SSCE}/\Delta\bar{T}$ under RCP8.5. **g**, Intermodel scatterplot between SSCF and the temperature-mediated precipitation increase under Abrupt4xCO₂. **h**, Intermodel scatterplot between Index C and SSCF. **i**, Intermodel scatterplot between Index C and the temperature-mediated precipitation increase under Abrupt4xCO₂. The open and filled dots are the CMIP5 and CMIP6 results, respectively.

Figure R5: Same as Fig. 2a-d but with SSCF diagnosed from Abrupt4xCO₂ as the x axis.

2. Regarding the evaporation calculations: can the authors expand on the ocean heat uptake (G) calculation a bit more in methods? If it is computed as a residual, this implies that it uses model-generated SW, LW, and E. How then is model-generated E different from the E computed from the Penman equation? Is one expected to be a better estimate than the other? Also, do the SW and LW terms appear twice or cancel out if one were to expand G in equation (2). In short, I wonder how strongly the model-generated and Penman E are dependent on each other (due to the way G is computed) and whether this dependence could explain the apparently good agreement between these quantities (e.g., Fig. S2).

The model-simulated E and the projected change are what the Penman equation tries to capture. The Penman equation, as a theoretical equation, allows us to understand how individual factors contribute to the model-projected E change and its intermodel uncertainty.

To clarify the computation of G, we would like to review the derivation of the Penman equation. Evaporation over open water can be written as,

$$L_v E = \kappa(e_s^* - e) = \kappa\Delta(T_s - T) + \kappa e^*(1 - RH),$$

where $\Delta \equiv \frac{\partial e^*}{\partial T} = \frac{L_v e^*}{R_v T^2}$ measures the slope of the $T - e^*$ curve and $\kappa \equiv \frac{\rho C_p}{\gamma r_a}$.

Given sensible heat flux $SH = \kappa\gamma(T_s - T)$, and surface energy balance $L_v E + SH = R_n - G$, Penman solves evaporation by eliminating $T_s - T$. That is,

$$L_v E = \frac{\Delta}{\Delta + \gamma}(R_n - G) + \frac{\gamma}{\Delta + \gamma}\kappa e^*(1 - RH),$$

where $R_n = SW + LW$ is the net surface radiation.

The Penman-estimated evaporation thus includes two parts: the first term in the R.H.S. represents the energy part which allocates $\frac{\Delta}{\Delta + \gamma}$ of total available energy ($R_n - G$) to E and the second term represents the aerodynamic part which reflects the effect of RH.

In model climatology (1979-2015), the global mean G is $\sim 1.34 \pm 0.8 \text{ W/m}^2$ while R_n is $\sim 115 \pm 3.5 \text{ W/m}^2$ (SSCE, $-59 \pm 5.5 \text{ W/m}^2$). Thus, G does not have a notable impact on the magnitude of global mean E and contributes secondarily to the intermodel spread. Because it is difficult to directly compute G , we diagnose G as $G = R_n - L_v E - SH$, so $R_n - G = L_v E + SH$, reflecting the total energy for surface turbulent heat flux. With G diagnosed, the Penman equation can be used to estimate how much the individual factors contribute to the intermodel spread in the E change. As shown in Fig. 1d, the increase in E is dominated by the increase in $\frac{\Delta}{\Delta + \gamma}$. This effect is independent of the change in $R_n - G$, i.e., $L_v E + SH$. The intermodel uncertainty in the E increase is dominated by SSCE and G plays a secondary role (Fig. 1d).

Hope this clarifies our computation of G and we have described the computation of G more clearly in Methods.

3. Has the sensitivity to the definition of C (the metric used for the emergent constraint) been explored? Currently, C depends on the relationship between GSSCF and climatological SSCE itself (Fig. 4a). Would a metric that is independent of this relationship (say, just the difference between climatological SSCE between the low and high latitudes) give the same tight relationships as in (b) and (c) of Fig. 4? Not only would this demonstrate robustness, but also allow one to compute the metric in other ensembles without the need to first generate Fig. 4a based on that ensemble.

Thank you for your suggestion! We have now introduced a simple index as the difference in the climatological SSCE between the region with observed annual-mean SST $> 25^\circ\text{C}$ and the region with observed annual-mean SST $> 0^\circ\text{C}$ but $< 25^\circ\text{C}$. As shown in Fig. R6, this simple index is significantly correlated with Index C , and is able to constrain the intermodel uncertainty of $\Delta \overline{SSCE} / \Delta \bar{T}$ and $\Delta P / P$. The resultant ECs indicate similar results as those of Index C , demonstrating the robustness of the proposed EC. We have included Fig. R6 as Fig. S9 and presented the related result at the end of the EC section.

Figure R6: Same as Fig. 4 but using the simple climatological index of SSCE, Index D. The red contour in a indicates the contour of 25°C.

Specific/minor comments:

L45-46: “a more recent analysis suggests that the statistical relationships underlying these two ECs do not hold up among the latest generation of climate models” – This claim is not entirely true. As stated in the abstract of ref. 21: “The constraint based on clear-sky shortwave absorption sensitivity to water vapor has weakened, and it is argued that a proposed constraint based on surface low cloud longwave radiative effects does not apply to HS.” Ref. 21 does not suggest that both ECs no longer apply or that all key related relationships no longer exist.

We have edited the sentence as follow: However, a more recent analysis²¹ finds that, in the latest generation of climate models, the correlation of HS to clear-sky shortwave feedback weakens substantially ($r = -0.45$) and the correlation to longwave cloud feedback no longer holds ($r = -0.01$).

L52-54: “The rationale is that future precipitation increase is correlated with future temperature increase and the latter is correlated with the forced trend over recent decades” – Did you mean to say “the former” rather than “the latter” is correlated...?

Sorry for the confusion. We mean “future precipitation increase should be correlated with future temperature increase and future temperature increase should be correlated with forced temperature trend over recent decades”. We have modified the sentence to clarify our point.

L54-55: “That said, only the CS part has been utilized in the EC to constrain DP/P while the potential to constrain the HS part has not been harnessed.” I'm not sure this is true since that study (ref. 22), according to its abstract, uses trends in DP itself to constrain future DP. It is thus constraining the total DP directly, which has contributions from HS and CS as the authors state earlier.

In Ref. 22, the correlation between future $\Delta P/P$ and recent ΔP is only 0.49, smaller than the correlation $r=0.6$ between future $\Delta P/P$ and recent ΔT . As a result, their paper mainly focused on the constraint using recent ΔT – future $\Delta P/P$. That said, the potential to constrain HS is not harnessed. The correlation between future $\Delta P/P$ and recent ΔP is mainly from the correlation between future $\Delta P/P$ and recent ΔT , and the correlation is small likely because precipitation trend is more variable. We have edited the paragraph to clarify these points.

Figs. 1, S2, and possible others: It would help to state the time periods that are plotted in the figure captions.

We have stated the time periods in the figure captions of Fig. 1 and Fig. S1 (but not other figures to avoid redundancy). The time periods are also stated in the last paragraph of Introduction and in Methods.

L126: I would avoid using such a long acronym (GSSCF) in the section title, especially since this acronym has not been defined yet.

We have edited the section title to “Surface shortwave cloud feedback regulates future precipitation change by affecting both CS and HS”.

L158-160: “The shortwave cloud feedback, widely considered as a major uncertainty source for CS, has been often studied from the perspective of the top of the atmosphere (TOA).” What is the relationship between the TOA and SFC SW cloud feedback? It is assumed here that these should be essentially the same, qualitatively and perhaps even quantitatively, to some extent? If so, this should be stated explicitly.

As shown in Fig. R7 (included in manuscript as Fig. S8), cloud SW effects/feedback are spatially correlated between TOA and SFC. The TOA SW cloud feedback can be constrained by its climatological pattern (Siler et al., 2018) and so is the SFC SW cloud feedback (this work). We have mentioned the correlation between TOA and SFC cloud shortwave effect in Line 183-184. **But we would like to note that, to build the constraint on SFC SW feedback, there is no need to first link SFC to TOA.**

Figure R7: The shortwave cloud effect and feedback at the TOA and SFC.

Response to Reviewer #3:

Several major revisions and comments are provided for the manuscript, as I found it unconvincing:

Abstract: what exactly is meant by climatological cloud effect? This is very vague and I don't know what exactly it is about clouds that is meant

Sorry for the confusion. We mean "the climatological pattern of cloud shortwave effect" and we have modified the text.

Line 27: Give a clear definition of what $\Delta P/P$ stands for (the fractional change in mean precip normalized by the mean precip)

We have edited the sentence to clearly describe $\Delta P/P$ as "the fractional increase in mean precipitation".

Line 33: "varying by a factor of 2" is unclear language.

We have edited the sentence to "varying by a factor of two among models (ratio between the highest and lowest values)".

Line 41: Emergent constraints should be constructed only with observable quantities, but this sentence seems to imply anything can be used?

It seems to us that we did not mention any unobservable quantities in this sentence (copied below).

"For CS, several ECs have been proposed from different perspectives⁸, such as linking future change in temperature to its seasonal⁹ or interannual variability¹⁰, relating cloud feedbacks to its climatological pattern^{11,12} and properties^{13,14}, or using correlations with other observable features^{15,16}."

The observable quantities here are the seasonal or interannual variability of temperature and the climatological pattern or properties of cloud effect, and other observable quantities. To construct ECs, one first identifies correlations between future projection

and some quantity in the model ensemble and then use the observations of that quantity to constrain model projection. Please let us know if we misunderstand this comment.

Line 48-49: “under different frameworks” and “different but likely dependent observables” are oddly worded and unclear.

We have modified the sentence to “It was challenging to do so as the ECs on HS and CS were based on different but potentially correlated observables”.

Line 51: Not sure this is necessary, seems like a rehash of previous paragraph? Also co-dependencies of terms within the EC as explained in previous paragraph?

The previous paragraph talks about ECs on HS ($\Delta P/P/\Delta T$) and CS (ΔT), while this paragraph talks about ECs that directly constrain $\Delta P/P$. We have edited both paragraphs to make our points clearer.

Line 59: All ECs should be physically guided, so this statement is unnecessary. That said, why a surface energy balance framework – what is new and of value here, as opposed to ECs based on the TOA budget, for example.

We have deleted “physically guided” and edited the sentences to highlight the surface perspective (Line 61-65). Furthermore, the benefit of the surface perspective is emphasized in Discussion.

Lines 36-57: The manuscript could be better motivated. It is good to explain what ECs are, but the focus is supposed to be on finding an EC for precipitation increases with warming, so why spend so much time discussing ECs for CS and HS? It’s distracting, focus instead on ECs for precipitation increase specifically. That said, I believe there has been other rather recent work in addition to Ref. 22 on ECs for precipitation increase, and the manuscript alludes to how this EC is different from other studies; this should be further discussed. What makes this framework for developing an EC in this manuscript new and of value? The motivation for using ocean evaporation and the surface energy budget is particularly missing.

Line 60-62: It's not necessary for an EC to constrain both HS and CS first in order to well constrain future precipitation increases (so long as a physical relationship exists between the observed quantity and the precipitation increase). This seems convoluted and potentially misguided. Why this approach?

These two comments are closely related, so we will answer them together.

$\Delta P/P$ can be written as $\Delta P/P = HS \times CS$. It may be possible to directly constrain $\Delta P/P$ without explicitly considering HS and CS. But we believe it is meaningful and helpful to attribute the uncertainty in $\Delta P/P$ to those in HS and CS.

First, as shown in Fig. R8, $\Delta P/P$ in individual models are scaled by ΔT , so it makes sense to first isolate the uncertainty from CS.

Figure R8: Intermodel scatterplot between ΔT and $\Delta P/P$.

Second, for our work, the EC works well to constrain $\Delta P/P$ because it constrains both HS and CS, so we would like to introduce our work from this perspective.

Finally, HS and CS are widely recognized parameters that have important physical meanings. HS is useful in itself for assessment of hydrological impact of climate change, because sometimes we would like to know the climate impact at specific degrees of

warming (e.g., 2°C) instead of under specific warming scenarios. For example, in IPCC AR6, both global warming levels and scenarios are used to discuss climate change.

Ref. 22 directly constrains $\Delta P/P$ by linking it to the historical temperature trend. This is based on the relations that $\Delta P/P$ is correlated with ΔT and ΔT is further correlated with forced temperature trend over recent decades. So, the EC is mainly based on the CS. They did try constraining $\Delta P/P$ by linking it to the historical precipitation trend, but the correlation is lower than that using the temperature trend. We found another study that provides some sort of constraint on $\Delta P/P$, but that was also based on the relation between $\Delta P/P$ and ΔT (Fig. 2 in Allen & Ingram, 2002).

Line 124: compensation from other factors maybe deserves to be explored – if we reduced uncertainty in SSCE then how much would it really matter given these other factors (ok, at least the discussion section acknowledges this, but this issue still needs to be addressed).

Thank you for your comment. As now shown in Fig. S5, we find that the intermodel uncertainty due to SSCE is partially compensated by that due to surface net longwave radiation. We have explicitly mentioned this compensation in Line 119-120.

We'd like to note that the effect of this compensation is already included in the regression line between $\overline{\Delta SSCE}/\Delta\bar{T}$ and $\overline{\Delta E}/\bar{E}/\Delta\bar{T}$ and consequently included in the resultant EC. Specifically, given the regression slope $\beta = \frac{\sum(x_i - \bar{x})(y_i - \bar{y})}{\sum(x_i - \bar{x})^2}$, the compensation reduces the regression slope between $\overline{\Delta SSCE}/\Delta\bar{T}$ and $\overline{\Delta E}/\bar{E}/\Delta\bar{T}$. The correlation between $\overline{\Delta SSCE}/\Delta\bar{T}$ and $\overline{\Delta E}/\bar{E}/\Delta\bar{T}$ should be less impacted, given $r = \frac{\sum(x_i - \bar{x})(y_i - \bar{y})}{\sqrt{\sum(x_i - \bar{x})^2 \sum(y_i - \bar{y})^2}}$ and $(y_i - \bar{y})$ is in both nominator and denominator. The correlation between $\overline{\Delta SSCE}/\Delta\bar{T}$ and $\overline{\Delta E}/\bar{E}/\Delta\bar{T}$ is $r=0.76$, so the uncertainty due to SSCE explains about 0.65 (i.e., $\sqrt{1 - r^2}$) of the uncertainty of $\overline{\Delta E}/\bar{E}/\Delta\bar{T}$.

Line 131: Calling the change in SSCE with warming a “cloud feedback” is misleading, since this includes feedbacks and adjustments but isn't quite itself a feedback, it's a

change in shortwave cloud radiative effects as seen by the surface. A different analysis is needed to isolate the surface shortwave cloud feedbacks.

Following your suggestion, we directly use the term $\Delta\overline{SSCE}/\Delta\bar{T}$ to refer to the change in SSCE with warming. Furthermore, we have computed the effect of rapid adjustment and temperature-mediated feedback based on the piControl and Abrupt4xCO2 simulations of the CMIP models (see details in Methods). We show that the intermodel uncertainty in $\Delta\overline{SSCE}/\Delta\bar{T}$ is dominated by the temperature-mediated feedback (Line 166-173).

Lines 145: There actually seems to a deterioration between the CMIP5 and CMIP6 ensembles in the correlation between GSSCF and change in evaporation and change in precipitation with warming, but especially for precipitation. The deterioration is fairly large for precipitation, and appears to be driven by two models. Which are these and why do they depart so starkly from the other CMIP6 models? Doesn't this negatively impact the EC constructed for precipitation, or is $r=0.67$ still a high enough correlation?

The correlation between GSSCF (now $\Delta\overline{SSCE}/\Delta\bar{T}$) and $\Delta\bar{E}_o/\bar{E}_o/\Delta T$ decreases from $r=0.8$ in CMIP5 to $r=0.74$ in CMIP6 (Fig. 2c) and the correlation between $\Delta\overline{SSCE}/\Delta\bar{T}$ and $\Delta\bar{P}/\bar{P}/\Delta T$ decreases from $r=0.82$ in CMIP5 to $r=0.67$ in CMIP6. The two models responsible for the decreased correlations in CMIP6 are CanESM5 and EC-Earth3. For CanESM5 (the model that departs from others in Fig. 3c), it projects a larger $\Delta\bar{E}_o/\bar{E}_o/\Delta T$ than that expected from $\Delta\overline{SSCE}/\Delta\bar{T}$. As shown in Fig. 1d, while $\Delta\overline{SSCE}/\Delta\bar{T}$ is the dominating factor for the uncertainty in $\Delta\bar{E}_o/\bar{E}_o/\Delta T$, other factors can also contribute. For EC-Earth3, its $\Delta\bar{E}_o/\bar{E}_o/\Delta T$ is as expected from its $\Delta\overline{SSCE}/\Delta\bar{T}$, but its $\Delta\bar{P}/\bar{P}/\Delta T$ is higher than expected. While we expect a close correlation between $\Delta\bar{E}_o/\bar{E}_o/\Delta T$ and $\Delta\bar{P}/\bar{P}/\Delta T$, they are not identical due to factors such as land processes.

If we remove these two models from the calculation, the correlation between $\Delta\overline{SSCE}/\Delta\bar{T}$ and $\Delta\bar{E}_o/\bar{E}_o/\Delta T$ increases from 0.74 to 0.82 in CMIP6 (from 0.76 to 0.81 in combined CMIP), and the correlation between $\Delta\overline{SSCE}/\Delta\bar{T}$ and $\Delta\bar{P}/\bar{P}/\Delta T$ increases from in CMIP6 from 0.67 to 0.78 (from 0.70 to 0.79 in combined CMIP).

Line 158: This approach seems odd – since GSSCF is correlated with $\Delta\bar{P}/\bar{P}$, why not just

find an observable variable that is highly correlated or linked to shortwave cloud feedback and build the EC for precipitation with that variable directly? That seems to me to be the standard way to construct an EC, rather than build an EC on top of an EC (constraining GSSCF to then constrain precipitation change), which may be a misguided approach. If this is what was done, this needs to be made more clear, because it does not read this way.

Thank you for your comment. We did not build an EC on top of another EC. We build an EC for $\overline{\Delta SSCE}/\Delta\bar{T}$ based on the correlation between Index C and $\overline{\Delta SSCE}/\Delta\bar{T}$ (Fig. 4b) and an EC for $\Delta P/P$ based on the correlation between Index C and $\Delta P/P$ (Fig. 4c). To avoid confusion, we have modified the sentence as “Now, we explore potential correlations of $\overline{\Delta SSCE}/\Delta\bar{T}$ to observable quantities and build ECs for $\overline{\Delta SSCE}/\Delta\bar{T}$ and $\Delta\bar{P}/\bar{P}$ ”.

Line 160: Need to make clear what understanding/advancement is gained by examining from a surface energy balance perspective rather than TOA.

Thank you for your suggestion. The surface-energy-balance perspective allows us to link the shortwave cloud effect to both HS and CS. The TOA perspective is usually used for understanding CS, but it does not offer an easy way to interpret the intermodel spread of HS. The advantage of the surface perspective is now emphasized in both Introduction and Discussion.

Line 162: Again, what exactly is meant by climatological cloud pattern? Just their geographical distribution, and/or optical thickness, etc.?

Sorry for the confusion. We mean the climatological pattern of cloud shortwave effect. We have edited the sentence.

Lines 167-168: Why ignoring the models in between these two groups?

The comparison between the two distinct groups (Fig. 3) is used for illustrating how intermodel uncertainty in $\overline{\Delta SSCE}/\Delta\bar{T}$ may be associated with the differences in the climatological SSCE. We later use regression (include all models, Fig. 4) to further show the dependency of $\overline{\Delta SSCE}/\Delta\bar{T}$ on the spatial pattern of the climatological SSCE.

Line 224: does EC break down for low emission scenario and why? How does this compare to ECs based on TOA framework?? Like in ref 22

For low emission scenario RCP2.6, the global mean temperature increases by $\sim 1^\circ\text{C}$ by 2100 and the global mean precipitation increases by $\sim 2\%$. It is challenging (and maybe not meaningful) to constrain the intermodel uncertainty as the natural variability becomes comparable to the forced change, and ECs are used to constrain the forced change due to model uncertainty.

We have discussed in detail how our EC is different from the EC in Ref. 22 (second paragraph in Discussion). Particularly, because our EC constrains both HS and CS and because our final observable is a climatological pattern, our EC reduces $\sim 25\%$ of the projection uncertainty in $\Delta P/P$, while the EC in Ref. 22 reduces only 13% (which corresponds to a $\sim 25\%$ reduction in variance, as reported in their paper).

Line 237: this wasn't shown for f , wasn't it the definition?? Also this wasn't shown or discussed at all really anyway

The energy allocation factor f was introduced in Line 87-89 and described in detail in Methods. It has also been used when we describe Fig. 1d. As derived in Methods, a warmer temperature will increase the energy allocation of total available energy to surface latent flux and contribute to future evaporation increase. We now use "the increasing energy allocation" in this line.

Fig. 4, remainder of EC construction, and Discussion: the EC presented in Fig. 4 seems veeeeery hand-wavy. The climatological SSCE index is heavily derived parameter, based on quantities that are observed but can have a high uncertainty, while ECs ideally should be constructed with observable parameters/variables that at the least are not so derived or directly measured, and the observable's uncertainty range tells something about the plausible range of the parameter being constrained by highlighting its values that fall into that range. I do not see that with the EC as presented in Fig. 4 for precipitation change with warming. The horizontal error bars in Fig. 4 appear to be missing in fact, so that this plausible range of precipitation change values does not seem like it can be easily

estimated despite the uncertainty value given in the text. It's really the horizontal error bars on the climatological index and NOT the vertical bars that we're after with an EC anyway, and so the uncertainty range given in the text appears to be misestimated; we want to see which values of the precipitation change are associated with the range of index values over its uncertainty range – this is the uncertainty range on the precipitation change rather than the calculation underlying the vertical error bars in Fig. 4. I remain unconvinced by this EC because of the amount of derivation required and the misguided uncertainty/constraint estimation.

In response to your concern, we have introduced a simpler Index D that measures the gradient of the climatological pattern of cloud shortwave effect. Specifically, the index is computed as the difference in the climatological SSCE between the region with observed annual-mean SST > 25°C and the region with observed annual-mean 0°C < SST < 25°C. Index D can be directly computed without pattern correlation or the exact knowledge of the regression map (Fig. 4a). As shown in Fig. S9 (copied here as Fig. R9), Index D effectively constrains $\Delta \overline{SSCE} / \Delta \bar{T}$ and $\Delta \bar{P} / \Delta \bar{P}$, and the resultant ECs indicate similar results as those of Index C. This demonstrates the robustness of the proposed EC and we have presented these results at the end of the EC section (Line 228-234).

For both Indices C and D, the uncertainty in observations is small, and particularly, much smaller than the uncertainty among models. Specifically, Index C is 0.28 in CERES and 0.25 in ISCCF but varies from -0.81 to 0.75 in models. Index D is 0.89 in CERES and 0.84 in ISCCF but varies from -0.99 to 1.94 in models. The error bars of the observations, as shown in Fig. 4 and Fig. S9, are thus very short. As a result, the uncertainty of the constrained $\Delta P / P$ is mainly from the residual variability that is unexplained by the correlation, instead of the uncertainty in observations. **This reflects the advantage of using a climatology as the final observable.** In Ref. 22, the final observable is the forced historical temperature trend, which includes substantial uncertainty itself. When computing the uncertainty range of the EC, we do include the effect of the uncertainty in observations (see Methods).

Figure R9: Same as Fig. 4 but using the simple climatological index of SSCE, Index D. The red contour in a indicates the contour of 25°C.

Minor Comments:

Fig. S4: what is the x-axis?? The model “number?” Yes, it is the model number. We have added this in the figure caption. Thank you for pointing this out.

REVIEWERS' COMMENTS

Reviewer #1 (Remarks to the Author):

Summary:

I thank the authors for the additional analyses which have well addressed my concerns and improved the robustness of the results. I have only a few minor suggestions.

1. Following my previous Major comment #1, the authors added a new section to discuss the difference between transient and quasi-stabilized stages under RCP4.5/SSP2-4.5 scenarios, which is helpful for understanding the hydrological responses under median emission scenarios. But in the text and figures, to be accurate, you should refer to “RCP4.5/SSP2-4.5 and RCP8.5/SSP5-8.5” rather than “RCP4.5 and RCP8.5”, because both CMIP5 and CMIP6 models are used.

2. Discussion section: The comparison between the ECs in this paper (based on ECs on both HS and CS) and in Ref. 22 (mainly based on EC on CS alone) has further implications. In this paper, the EC based on both HS and CS leads to a larger $\Delta P/P$ after constraint; while the EC based on CS alone leads to a smaller $\Delta P/P$ after constraint in Ref. 22. The difference between the two methods might imply that for the projection uncertainty of precipitation change, even though CS exhibits a significant correlation with $\Delta P/P$ across models, CS alone may not be sufficient to constrain the inter-model uncertainty of precipitation projections, because other processes related to precipitation change are also important. I think this point is worth to be mentioned. In a broader sense, this actually indicates that one needs to be more careful when establishing emergent constraints.

Reviewer #2 (Remarks to the Author):

The authors have made substantial improvements to the manuscript and have addressed my previous major concerns. In particular, the assessment of the Abrupt4xCO₂ experiments helps with the interpretation of the physical origin of the emergent constraints. The authors also show the robustness of their results by demonstrating that a simpler observable metric leads to the same conclusions. I have only a few remaining minor comments:

1. I recommend being careful when referring to the temperature-mediated surface SW cloud radiative effect (SSCE) response to warming (DSSCE/DT) as the surface shortwave cloud feedback, as the term “feedback” in this context usually refers to the SW cloud radiative effect that is corrected for non-cloud factors (or could be masking, e.g., Soden et al. 2008, <https://doi.org/10.1175/2007JCLI2110.1>). I thus suggest avoiding using the word “feedback” as it pertains to clouds and explicitly saying “temperature-mediated component of DSSCE/DT” or something similar. Though it will add more words, it will avoid misinterpretation of what is being calculated.

2. In Methods, I suggest listing the models in a table rather than embedding the model names in a paragraph as currently written.

3. L206-208: rather than saying “A higher DSSCE/DT is associated with lower climatological SSCE over the warm tropical regions and higher climatological SSCE over cold midlatitudinal regions,” I suggest replacing “lower” with “less negative,” and “higher” with “more negative.”

4. There are a number of typos and grammatical errors throughout the study, many likely introduced during the latest round of revisions. I suggest the authors thoroughly proofread and edit the text. Below are a few examples, though the following list is not exhaustive:

L111: Change “they are the largest...” to “it is the largest...”

L157, Fig. 2 caption: “in term of” should be “in terms of”

Fig. S7 caption: “Abrupt4xCO2” -> “Abrupt4xCO₂”

L290: “we compares” -> “we compare”

L344 in Methods: change “hold fixed” to “held fixed”

Reviewer #3 (Remarks to the Author):

The authors' responses to my comments and the other reviewer's comments and their changes to their manuscript are satisfactory.

Response to Reviewer #1:

Summary:

I thank the authors for the additional analyses which have well addressed my concerns and improved the robustness of the results. I have only a few minor suggestions.

1. Following my previous Major comment #1, the authors added a new section to discuss the difference between transient and quasi-stabilized stages under RCP4.5/SSP2-4.5 scenarios, which is helpful for understanding the hydrological responses under median emission scenarios. But in the text and figures, to be accurate, you should refer to “RCP4.5/SSP2-4.5 and RCP8.5/SSP5-8.5” rather than “RCP4.5 and RCP8.5”, because both CMIP5 and CMIP6 models are used.

Following your suggestion, we have replaced “RCP4.5 and RCP8.5” with “RCP4.5/SSP2-4.5 and RCP8.5/SSP5-8.5” throughout the manuscript and Supplementary Information.

2. Discussion section: The comparison between the ECs in this paper (based on ECs on both HS and CS) and in Ref. 22 (mainly based on EC on CS alone) has further implications. In this paper, the EC based on both HS and CS leads to a larger $\Delta P/P$ after constraint; while the EC based on CS alone leads to a smaller $\Delta P/P$ after constraint in Ref. 22. The difference between the two methods might imply that for the projection uncertainty of precipitation change, even though CS exhibits a significant correlation with $\Delta P/P$ across models, CS alone may not be sufficient to constrain the inter-model uncertainty of precipitation projections, because other processes related to precipitation change are also important. I think this point is worth to be mentioned. In a broader sense, this actually indicates that one needs to be more careful when establishing emergent constraints.

Thank you for your suggestion. We have highlighted in the second point that “our EC utilizes both the HS and CS components to constrain $\Delta \bar{P}/\bar{P}$ while the EC in ref. (Shiogama et al., 2022) mainly utilizes the CS component”. At the end of this discussion, we have added the following sentence: “Caution is thus needed when interpreting the EC based solely on the observed temperature trend.”

Response to Reviewer #2:

The authors have made substantial improvements to the manuscript and have addressed my previous major concerns. In particular, the assessment of the Abrupt4xCO2 experiments helps with the interpretation of the physical origin of the emergent constraints. The authors also show the robustness of their results by demonstrating that a simpler observable metric leads to the same conclusions. I have only a few remaining minor comments:

1. I recommend being careful when referring to the temperature-mediated surface SW cloud radiative effect (SSCE) response to warming (DSSCE/DT) as the surface shortwave cloud feedback, as the term “feedback” in this context usually refers to the SW cloud radiative effect that is corrected for non-cloud factors (or could be masking, e.g., Soden et al. 2008, <https://doi.org/10.1175/2007JCLI2110.1>). I thus suggest avoiding using the word “feedback” as it pertains to clouds and explicitly saying “temperature-mediated component of DSSCE/DT” or something similar. Though it will add more words, it will avoid misinterpretation of what is being calculated.

Thank you for your suggestion. We understand that, when quantifying climate feedbacks using the radiative-kernel method (e.g., Soden et al., 2008), cloud feedback is estimated from <the change in the cloud radiative effect> **minus** <the effect of clouds on the TOA radiation anomaly induced by the changes in the non-cloud factors>. However, we also notice that in other studies of cloud effects, “feedback” is used generally to refer to the sensitivity of the total change in cloud radiative effect to temperature (e.g., Zelinka et al., 2017; Siler et al., 2018; Watanabe et al., 2018; Ceppi & Nowack, 2021).

In the manuscript, we have used the terms “the temperature-mediated feedback (of SSCE)” and “surface shortwave cloud feedback” but avoided the exact short term “cloud feedback”. We feel that replacing all “feedback” to “the temperature-mediated component of $\Delta\overline{SSCE}/\Delta\bar{T}$ ” would affect the readability of some sentences, so after some consideration we choose to keep the “feedback” term but clarify its definition when we first introduce it (Line 169-172):

“ $\Delta\overline{SSCE}/\Delta\bar{T}$ is the change in global ocean SSCE per unit warming under the RCP8.5/SSP5-8.5 scenario, which includes the effects of both rapid adjustment and temperature-mediated feedback (feedback here refers to the temperature-mediated change in SSCE, without corrections for non-cloud factors as in the radiative-kernel method).”

- Ceppi, P., & Nowack, P. (2021). Observational evidence that cloud feedback amplifies global warming. *Proceedings of the National Academy of Sciences*, 118(30), e2026290118. <https://doi.org/10.1073/pnas.2026290118>
- Shiogama, H., Watanabe, M., Kim, H., & Hirota, N. (2022). Emergent constraints on future precipitation changes. *Nature*, 602(7898), Article 7898. <https://doi.org/10.1038/s41586-021-04310-8>
- Siler, N., Po-Chedley, S., & Bretherton, C. S. (2018). Variability in modeled cloud feedback tied to differences in the climatological spatial pattern of clouds. *Climate Dynamics*, 50(3), 1209–1220. <https://doi.org/10.1007/s00382-017-3673-2>
- Watanabe, M., Kamae, Y., Shiogama, H., DeAngelis, A. M., & Suzuki, K. (2018). Low clouds link equilibrium climate sensitivity to hydrological sensitivity. *Nature Climate Change*, 8(10), Article 10. <https://doi.org/10.1038/s41558-018-0272-0>
- Zelinka, M. D., Randall, D. A., Webb, M. J., & Klein, S. A. (2017). Clearing clouds of uncertainty. *Nature Climate Change*, 7(10), Article 10. <https://doi.org/10.1038/nclimate3402>

2. In Methods, I suggest listing the models in a table rather than embedding the model names in a paragraph as currently written.

Following your suggestion, we have listed the models using a table (Table S1).

3. L216-218: rather than saying “A higher DSSCE/DT is associated with lower climatological SSCE over the warm tropical regions and higher climatological SSCE over cold midlatitudinal regions,” I suggest replacing “lower” with “less negative,” and “higher” with “more negative.”

We have edited the sentence following your suggestion.

4. There are a number of typos and grammatical errors throughout the study, many likely introduced during the latest round of revisions. I suggest the authors thoroughly proofread and edit the text. Below are a few examples, though the following list is not exhaustive:

L111: Change “they are the largest...” to “it is the largest...” **Corrected**

L157, Fig. 2 caption: “in term of” should be “in terms of” **Corrected**

Fig. S7 caption: “Abript4xCO2” -> “Abrupt4xCO2” **Corrected**

L290: “we compares” -> “we compare” **Corrected**

L344 in Methods: change “hold fixed” to “held fixed” **Corrected**

We have corrected the typos above and proofread the text carefully. Thank you!